# EMMA-X: An EM-like Multilingual Pre-training Algorithm for Cross-lingual Representation Learning

Ping Guo[‡,§]    Xiangpeng Wei[†,*]    Yue Hu[‡,§,*]
Baosong Yang[†]    Dayiheng Liu[†]    Fei Huang[†]    Jun Xie[†]

[‡]Institute of Information Engineering, Chinese Academy of Sciences, Beijing, China
[§]School of Cyber Security, University of Chinese Academy of Sciences, Beijing, China
[†]Machine Intelligence Technology Lab, Alibaba DAMO Academy, Hangzhou, China
{guoping, huyue}@iie.ac.cn, pemywei@gmail.com

## Abstract

Expressing universal semantics common to all languages is helpful in understanding the meanings of complex and culture-specific sentences. The research theme underlying this scenario focuses on learning universal representations across languages with the usage of massive parallel corpora. However, due to the sparsity and scarcity of parallel data, there is still a big challenge in learning authentic "universals" for any two languages. In this paper, we propose EMMA-X: an **EM**-like **M**ultilingual pre-training **A**lgorithm, to learn **(X)C**ross-lingual universals with the aid of excessive multilingual non-parallel data. EMMA-X unifies the cross-lingual representation learning task and an extra semantic relation prediction task within an EM framework. Both the extra semantic classifier and the cross-lingual sentence encoder approximate the semantic relation of two sentences, and supervise each other until convergence. To evaluate EMMA-X, we conduct experiments on **XRETE**, a newly introduced benchmark containing 12 widely studied cross-lingual tasks that fully depend on sentence-level representations. Results reveal that EMMA-X achieves state-of-the-art performance. Further geometric analysis of the built representation space with three requirements demonstrates the superiority of EMMA-X over advanced models [2].

## 1 Introduction

Research on how to express universal semantics for natural languages (metaphorically as "alphabet of human thoughts" by Leibniz and von Leibniz [1996]) has lasted a long time. Usually, these universal meanings underlying all human natural languages are referred to as irreducible semantic cores [Wierzbicka, 1999]. These common cores across languages can serve as a bridge, to help better understand the exact meanings of complex sentences in different languages.

In the context of computational linguistics, various works [Huang et al., 2019, Conneau et al., 2020, Chi et al., 2021, Wei et al., 2021, Lee et al., 2022, Li et al., 2023, Chen et al., 2023] have led to great improvements on learning cross-lingual universal representations with the usage of parallel corpora, and verify that multilingual universality contributes a major performance on cross-lingual understanding. However, due to the sparsity and scarcity of parallel data, these advanced techniques face a big challenge in learning real universality among all languages. For instance, among the widely-available top 100 languages that theoretically can build 4950 language pairs, only about 200 language pairs have considerable parallel data [Aharoni et al., 2019, Bapna et al.,

---

[*]Corresponding Author.
[2]Codes and datasets of the XRETE benchmark: https://github.com/guopingiie/EMMA-X

37th Conference on Neural Information Processing Systems (NeurIPS 2023).

2022]. Recently, Large Language Models (LLMs) (e.g., PaLM [Chowdhery et al., 2022], OPT [Zhang et al., 2022b], BLOOMZ [Workshop et al., 2023], ChatGPT, etc.) have reached a milestone in the field of Natural Language Processing, for their promising capability at understanding and following complex natural language instructions in different languages. By modeling a wide variety of sentence samples in discrete sentence space, LLMs can capture some universal linguistic phenomena to gain cross-lingual transferability. This is consistent with our goal of building a universal basement that supports all languages. The difference lies in that we achieve it through learning universal continuous representations across different languages.

Concretely, we propose EMMA-X to tackle the above challenge from a continuous perspective. EMMA-X can learn cross-lingual universal sentence representations with excessive non-parallel multilingual data by unifying two highly dependent tasks in an EM [Moon, 1996] framework: semantic relation classification and cross-lingual universal representation learning. For the former, we introduce a Gaussian Mixture Model [Everitt and Hand, 1981] classifier (**GMM classifier**) to deal with the key challenge of forming positive sentence pairs for non-parallel multilingual corpora, by annotating the semantic relationship of sentence pairs in any two arbitrary languages on the fly. For the latter, we employ **a cross-lingual encoder** to learn universal sentence representations via contrastive learning, where positive pairs are chosen by GMM classifier. Further, we construct training signals according to the output of the cross-lingual encoder, to inversely supervise GMM classifier. From the perspective of EM algorithm, in E-step, both modules try to approximate the semantic relationship given a sentence pair sampled from two arbitrary languages. One module is supervised by the approximation of the other to build its own expectation. In M-step, two modules update their parameters by maximizing expectations, respectively. We give a theoretical justification about how these two tasks can be interpreted from an EM perspective (Section 4).

To incentivize the research of universal sentence representation learning, we form a Cross-lingual REpresentation Transfer Evaluation (**XRETE**) benchmark, which includes 12 cross-lingual tasks covering more than 50 languages. XRETE fully depends on sentence-level representations. Experimental results demonstrate that EMMA-X significantly outperforms pre-trained language models [Conneau et al., 2020, Chi et al., 2021] by 32% at most on XRETE. We also perform an evaluation of ChatGPT on XRETE to explore its multilingual performance. Detailed analysis also shows that EMMA-X can mitigate the representation discrepancy between head and massive long-tail languages. We further conduct geometric analysis directly on representation space from three perspectives: Invariance [Abend and Rappoport, 2017], Canonical Form [Teller, 2000] and Isotropy [Mu and Viswanath, 2018], which provides a further understanding of the cross-lingual transferability of these models.

## 2  Preliminaries

Cross-lingual representation learning aims at mapping sentences from different languages into a unified continuous space, where synonyms across different languages are pulled closer. Given a sentence $\mathbf{x}$, the representation is formulated as

$$\gamma^{(\mathbf{x})} = f\big[g\big(\mathcal{M}(\mathbf{x}; \Theta_{\mathcal{M}})\big)\big], \tag{1}$$

where $\mathcal{M}(\cdot; \Theta_{\mathcal{M}})$ denotes the encoder network with a set of trainable parameters $\Theta_{\mathcal{M}}$, which is typically implemented as a transformer encoder architecture [Vaswani et al., 2017, Devlin et al., 2019, Lee et al., 2022, Feng et al., 2022]. $f(\cdot)$ is L-2 normalization and $g(\cdot)$ is the aggregate function. We take the final hidden states of "[CLS]" token as the aggregate sentence representation.

To learn reasonable representations that can express universal semantics across different languages, various well-designed techniques have been applied to $\gamma^{(\mathbf{x})}$. A predominant one is to build contrastive learning (CTL) [Saunshi et al., 2019] objective with parallel corpora. The basic idea is to maximize the similarity between representations (i.e., $\gamma^{(\mathbf{x})}$ and $\gamma^{(\mathbf{y})}$) of two semantically-equivalent sentences $(\mathbf{x}, \mathbf{y})$, while keep randomly sampled irrelevant ones $\gamma^{(\mathbf{y}')}$ away. Formally, assume $\mathcal{B}$ to be a batch of multilingual parallel bitexts, the contrastive loss under InfoNCE [Oord et al., 2018] formulation is

$$\mathcal{L}_{\mathbf{CTL}} = -\log \frac{e^{s(\gamma^{(\mathbf{x})}, \gamma^{(\mathbf{y})})}}{e^{s(\gamma^{(\mathbf{x})}, \gamma^{(\mathbf{y})})} + \sum_{\mathbf{y}' \in \mathcal{B}, \mathbf{y}' \neq \mathbf{y}} e^{s(\gamma^{(\mathbf{x})}, \gamma^{(\mathbf{y}')})}}, \tag{2}$$

where $s(\cdot)$ is implemented as the cosine similarity $s(\gamma^{(\mathbf{x})}, \gamma^{(\mathbf{y})}) = \frac{\gamma^{(\mathbf{x})\top}\gamma^{(\mathbf{y})}}{\|\gamma^{(\mathbf{x})}\| \cdot \|\gamma^{(\mathbf{y})}\|}$, $\mathbf{y}$ and $\mathbf{y}'$ are typically called positive and negative samples.

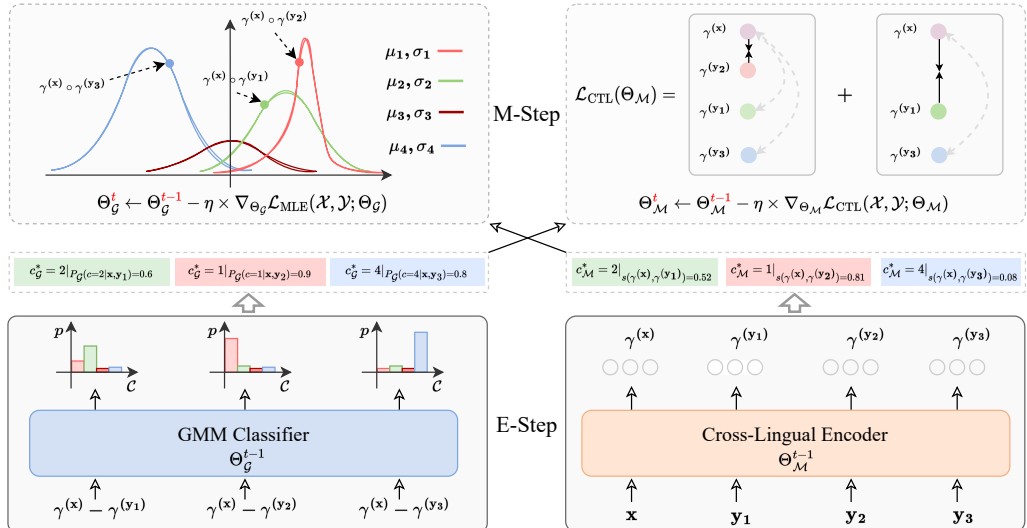

Figure 1: Illustration of EMMA-X, involving two modules (GMM classifier and Cross-lingual Encoder) that are reciprocated from each other and are updated alternatively. $\mathbf{x}$ means the current instance, $\{\mathbf{y}_1, \mathbf{y}_2, \mathbf{y}_3, ...\}$ are samples in various languages for comparison. $\gamma^{(\mathbf{x})}$ is the continuous representation given a discrete sentence $\mathbf{x}$. $c_{\mathcal{M}}^*$ and $c_{\mathcal{G}}^*$ formulate the semantic ranks approximated according to Eq. 10 and Eq. 9, to supervise the GMM classifier and cross-lingual encoder, respectively.

## 3 Methodology

We propose EMMA-X that fully exploits massive monolingual data to learn cross-lingual universal representations. As illustrated in Figure 1, EMMA-X consists of two modules: 1) A GMM classifier $\mathcal{G}(\cdot; \Theta_{\mathcal{G}})$ to approximate the semantic relation of non-parallel sentences. 2) A cross-lingual encoder $\mathcal{M}(\cdot; \Theta_{\mathcal{M}})$ to convert multilingual sentences into universal representations. For optimization, EMMA-X unifies these two modules in an EM framework with dual supervision. In this section, we begin with a definition of the semantic relation rank (§3.1). Then, we introduce model initialization (§3.2) and the proposed training paradigm (§3.3), followed by a dual supervision strategy (§3.4). For a clearer presentation, an Algorithm of EMMA-X is shown in Algorithm 1.

### 3.1 Rank of Semantic Relations

Mainstream methods model semantic relations with a strict binary separation: positives and negatives. However, the boundary between positives and negatives is blurry, and many samples cannot be clearly classified as either positives or negatives. So it cannot maximize the potential of models to perceive more subtle semantic changes. Also, a binary separation will lead to far more negative samples than positive ones (imbalanced data). To more accurately capture the semantic relation between two sentences and alleviate imbalanced problem, we subdivide the relation into $N$ semantic ranks, where the semantic similarity of each rank decreases as $N$ increases, e.g., $c = 1$ denotes two sentences are paraphrases of each other, while $c = N$ implies sentences are irrelevant. In practice, we set $N$ to 4.

### 3.2 Model Initialization

In EMMA-X, the GMM classifier $\mathcal{G}(\cdot; \Theta_{\mathcal{G}})$ and cross-lingual encoder $\mathcal{M}(\cdot; \Theta_{\mathcal{M}})$ are initialized by training with massive parallel corpora, respectively.

**Initialization of Cross-lingual Encoder.** It is initialized with XLM-R [Conneau et al., 2020] and then continuously trained with InfoNCE [Oord et al., 2018] loss by Eq.2. Following HICTL [Wei et al., 2021] and INFOXLM [Chi et al., 2021], we treat the parallel sentence pairs as the query sentence $\mathbf{x}$ and its positive counterpart $\mathbf{y}$, while treating the randomly sampled sentence as a negative one $\mathbf{y}'$.

**Initialization of GMM Classifier.** A reasonable solution to warm up the GMM classifier is to use the available cross-lingual parallel corpora as training signals. Suppose $\mathbf{x}$ and $\mathbf{y}$ are parallel sentences,

and $\mathbf{y}'$ is an outlier. We set the semantic ranks for $(\gamma^{(\mathbf{x})}, \gamma^{(\mathbf{y})})$ and $(\gamma^{(\mathbf{x})}, \gamma^{(\mathbf{y}')})$ as $c = 1$ and $c = N$, respectively, according to the definition described in § 3.1. To obtain the fine-grained semantic ranks, we design a linear interpolation strategy similar to Wei et al. [2022] and mixup [Zhang et al., 2018], which provides virtual training examples for each semantic rank. Formally,

$$\boldsymbol{\gamma}^{(\tilde{\mathbf{y}})} = (1 - \lambda) \cdot \boldsymbol{\gamma}^{(\mathbf{y})} + \lambda \cdot \boldsymbol{\gamma}^{(\mathbf{y}')}, \tag{3}$$

where $\lambda \in [0, 1]$ is sampled from a uniform distribution. We compute $r = \lceil (1 - \lambda) \cdot (c = 1) + \lambda \cdot (c = N) \rceil$ as the soft semantic rank for $(\gamma^{(\mathbf{x})}, \gamma^{(\tilde{\mathbf{y}})})$, where $\lceil \cdot \rceil$ means the least integer greater than or equal to the input. The virtual training examples are grouped together with the real parallel corpora to optimize the GMM classifier:

$$\mathcal{L}_{\mathbf{MLE}} = -\log P_{\mathcal{G}}(c = 1 | \mathbf{x}, \mathbf{y}; \Theta_{\mathcal{G}}) - \log P_{\mathcal{G}}(c = N | \mathbf{x}, \mathbf{y}'; \Theta_{\mathcal{G}}) - \sum_{\tilde{\mathbf{y}}} \log P_{\mathcal{G}}(c = r | \mathbf{x}, \tilde{\mathbf{y}}; \Theta_{\mathcal{G}}). \tag{4}$$

The posterior probability $P_{\mathcal{G}}(\cdot)$ is formulated as

$$P_{\mathcal{G}}(c = r | \mathbf{x}, \mathbf{y}; \Theta_{\mathcal{G}}) = \frac{\pi_r \cdot \mathcal{N}_r\big(\gamma^{(\mathbf{x})} - \gamma^{(\mathbf{y})} | \mu_r, \sigma_r\big)}{\sum_{j=1}^{N} \pi_j \cdot \mathcal{N}_j\big(\gamma^{(\mathbf{x})} - \gamma^{(\mathbf{y})} | \mu_j, \sigma_j\big)}, \tag{5}$$

where the Gaussian form $\mathcal{N}_r$ is assigned with a prior probability $\pi_r$, mean $\mu_r$ and standard deviation $\sigma_r$ that are all parameterized by trainable variables, thus $\Theta_{\mathcal{G}} = \{(\pi_r, \mu_r, \sigma_r) \mid r \in [1, N]\}$. See **Appendix G** for the specific calculation of Gaussian form $\mathcal{N}_r$.

### 3.3 The EM Iteration Framework

After initialization, EMMA-X further trains the GMM classifier $\mathcal{G}(\cdot; \Theta_{\mathcal{G}})$ and cross-lingual encoder $\mathcal{M}(\cdot; \Theta_{\mathcal{M}})$ with only multilingual non-parallel data with an EM framework.

**E-Step.** For optimization, we represent a training batch of multilingual non-parallel sentences as $\mathcal{X} = \{\mathbf{x}_1, \mathbf{x}_2, ..., \mathbf{x}_I\}$ accompanied by a queue of random sentences as $\mathcal{Y} = \{\mathbf{y}_1, \mathbf{y}_2, ..., \mathbf{y}_K\}$ for instance comparison. Formally, the expectation for GMM classifier is:

$$\mathcal{L}_{\mathbf{MLE}}(\mathcal{X}, \mathcal{Y}; \Theta_{\mathcal{G}}) = -\mathbb{E}_{\mathbf{x}_i \sim \mathcal{X}} \mathbb{E}_{\mathbf{y}_k \sim \mathcal{Y}} \big[ \log P_{\mathcal{G}}(c = c_{\mathcal{M}}^* | \mathbf{x}_i, \mathbf{y}_k; \Theta_{\mathcal{G}}) \big], \tag{6}$$

where $c_{\mathcal{M}}^* \in [1, N]$ represents an approximated semantic rank for the combination of anchor $\mathbf{x}_i$ and another random sentence $\mathbf{y}_k$, based on the cosine similarity among representations (i.e., $\gamma^{(\mathbf{x}_i)}$ and $\gamma^{(\mathbf{y}_k)}$) produced by the cross-lingual encoder (i.e., $\mathcal{M}(\cdot; \Theta_{\mathcal{M}})$). Please refer to §3.4 for details.

Correspondingly, the expectation for the cross-lingual encoder can be calculated with contrastive learning, where the positive samples are established by the maximum a posteriori approximation (argmax prediction) $c_{\mathcal{G}}^*$ given by the GMM classifier. Specifically, we apply ranking InfoNCE [Hoffmann et al., 2022] as the training objective, which recursively takes parallel sentence pairs in each rank (e.g, $c_{\mathcal{G}}^*$) as positives and ranks that are larger than $c_{\mathcal{G}}^*$ as negatives. Formally,

$$\mathcal{L}_{\mathbf{CTL}}(\mathcal{X}, \mathcal{Y}; \Theta_{\mathcal{M}}) = -\mathbb{E}_{\mathbf{x}_i \sim \mathcal{X}} \Bigg[ \log \frac{\sum_{\mathbf{y}_k \sim \mathcal{Y}_{c_{\mathcal{G}}^*=1}} e^{s[\gamma^{(\mathbf{x}_i)}, \gamma^{(\mathbf{y}_k)}]}}{\sum_{\mathbf{y}_t \sim \mathcal{Y}_{c_{\mathcal{G}}^* \in [1, N]}} e^{s[\gamma^{(\mathbf{x}_i)}, \gamma^{(\mathbf{y}_t)}]}} + \log \frac{\sum_{\mathbf{y}_k \sim \mathcal{Y}_{c_{\mathcal{G}}^*=2}} e^{s[\gamma^{(\mathbf{x}_i)}, \gamma^{(\mathbf{y}_k)}]}}{\sum_{\mathbf{y}_t \sim \mathcal{Y}_{c_{\mathcal{G}}^* \in [2, N]}} e^{s[\gamma^{(\mathbf{x}_i)}, \gamma^{(\mathbf{y}_t)}]}} + ... + \log \frac{\sum_{\mathbf{y}_k \sim \mathcal{Y}_{c_{\mathcal{G}}^*=N-1}} e^{s[\gamma^{(\mathbf{x}_i)}, \gamma^{(\mathbf{y}_k)}]}}{\sum_{\mathbf{y}_t \sim \mathcal{Y}_{c_{\mathcal{G}}^* \in [N-1, N]}} e^{s[\gamma^{(\mathbf{x}_i)}, \gamma^{(\mathbf{y}_t)}]}} \Bigg], \tag{7}$$

where $c_{\mathcal{G}}^* \in [1, N]$ represents a semantic rank approximated by the posteriori of GMM classifier (§3.4). For simplicity, we omit the temperature term in Eq. 7, and please see **Appendix G** for details.

**M-Step.** We use gradient descent algorithm to update the parameters of each module by optimizing its expectation. At each time step $t$, where $\eta$ and $\eta'$ are learning rates for each expectation, formally,

$$\begin{aligned} \Theta_{\mathcal{G}}^{t+1} &\leftarrow \Theta_{\mathcal{G}}^t - \eta \times \nabla_{\Theta_{\mathcal{G}}} \mathcal{L}_{\mathbf{MLE}}(\mathcal{X}, \mathcal{Y}; \Theta_{\mathcal{G}}), \\ \Theta_{\mathcal{M}}^{t+1} &\leftarrow \Theta_{\mathcal{M}}^t - \eta' \times \nabla_{\Theta_{\mathcal{M}}} \mathcal{L}_{\mathbf{CTL}}(\mathcal{X}, \mathcal{Y}; \Theta_{\mathcal{M}}). \end{aligned} \tag{8}$$

---

**Algorithm 1:** EMMA-X Training Algorithm

---

**Input:** multilingual parallel and non-parallel corpora
**Output:** $\mathcal{M}(\cdot; \Theta_{\mathcal{M}})$ and $\mathcal{G}(\cdot; \Theta_{\mathcal{G}})$
Phase 1 ;                  /* Warm-up two modules with multilingual parallel corpora */
**while** *not convergence* **do**
    Sample a batch of multilingual bitexts $\mathcal{B}$;
    **for** $(\mathbf{x}, \mathbf{y}) \in \mathcal{B}$ **do**
        Sample $\mathbf{y}' \in \mathcal{B}, \mathbf{y}' \neq \mathbf{y}$, Build virtual training examples $\tilde{\mathbf{y}}$ with Eq. 3;
        Compute $\mathcal{L}_{\mathbf{CTL}}$ with Eq. 2 to update $\Theta_{\mathcal{M}}$ and $\mathcal{L}_{\mathbf{MLE}}$ with Eq. 4 to update $\Theta_{\mathcal{G}}$;
    **end**
**end**
Phase 2 ;    /* EMMA-X training with EM framework using only non-parallel corpora */
**while** *not convergence* **do**
    **E-Step**
        Sample a batch of multilingual anchors $\mathcal{X}$, Queue a batch of random sentences $\mathcal{Y}$;
        **for** $(\mathbf{x}_i, \mathbf{y}_k) \in \mathcal{X} \times \mathcal{Y}$ **do**
            Approximate semantic rank $c_{\mathcal{G}}^*$ with Eq. 9 and $c_{\mathcal{M}}^*$ with Eq. 10 ;
        **end**
        Compute $\mathcal{L}_{\mathbf{MLE}}(\mathcal{X}, \mathcal{Y}; \Theta_{\mathcal{G}})$ according to Eq. 6 and $\mathcal{L}_{\mathbf{CTL}}(\mathcal{X}, \mathcal{Y}; \Theta_{\mathcal{M}})$ according to Eq. 7;
    **M-Step**
        Update $\Theta_{\mathcal{M}}$ and $\Theta_{\mathcal{G}}$ according to Eq. 8;
**end**

---

## 3.4 Dual Supervision

The approximated semantic ranks $c_{\mathcal{G}}^*$ and $c_{\mathcal{M}}^*$ are critical in EMMA-X training algorithm. To preserve their quality, we propose `dual supervision`: predictions from one module are fed to the other to calculate the expectation. In this section, we explain in detail how we approximate the semantic ranks for GMM classifier and cross-lingual encoder, respectively.

**Approximate Semantic Rank with GMM classifier.** The way to obtain semantic rank with semantic classifier is straightforward. The semantic rank corresponding to the highest probability among multiple Gaussian distributions is chosen as the prediction, which is further used to supervise the cross-lingual encoder $\mathcal{M}(\cdot; \Theta_{\mathcal{M}})$, as illustrated in Eq. 7. Formally,

$$c_{\mathcal{G}}^* = \underset{r}{\operatorname{argmax}} \, P_{\mathcal{G}}(c = r | \mathbf{x}_i, \mathbf{y}_k; \Theta_{\mathcal{G}}). \tag{9}$$

**Approximate Semantic Rank with Cross-lingual Encoder.** One common way to calculate sentence relation is to measure the similarity between two real-valued representations. Assuming $s_r$ (a scalar initialized as $\frac{r}{N}$) can reflect the general similarity score in semantic rank $c = r$. Given a random sentence pair $(\mathbf{x}_i, \mathbf{y}_k)$, if its similarity score is close to $s_r$, the sentence pair is likely to belong to rank $c = r$. Cross-lingual encoder $\mathcal{M}(\cdot; \Theta_{\mathcal{M}})$ determines the semantic relation for each pair according to

$$c_{\mathcal{M}}^* = \underset{r}{\operatorname{argmin}} \, |s(\gamma^{(\mathbf{x}_i)}, \gamma^{(\mathbf{y}_k)}) - s_r|, \tag{10}$$

where $|\cdot|$ refers to absolute value. Symmetrically, $c_{\mathcal{M}}^*$ is used to supervise GMM classifier (Eq. 6).

During the training process, the general similarity score for each semantic rank may vary. Thus, we propose a moving-average strategy to adaptively adjust the value of $s_r$ to simulate this change. Specifically, at time step $t$, $s_r$ is updated by cosine similarity of all the sentence pairs, which are currently categorized into the rank $c = r$ according to the cross-lingual encoder in Eq. 9.

$$s_r^t \leftarrow \epsilon \cdot s_r^{t-1} + (1 - \epsilon) \cdot s(\gamma^{(\mathbf{x}_i)}, \gamma^{(\mathbf{y}_k)}), \qquad \text{if } \mathbf{y}_k \in \mathcal{Y}_{c_{\mathcal{G}}^* = r}. \tag{11}$$

Here $\epsilon \in [0, 1]$ is a momentum coefficient to make $s_n$ evolve smoothly during training.

## 4 Theoretical Analysis

In this section, we provide theoretical justification for EMMA-X and demonstrate the mutual influence between two modules with rigorous interpretation from an EM algorithm perspective. We show that

under dual supervision, minimizing the positive terms in Eq. 7 intrinsically maximizes the objective of a classical clustering algorithm. For simplicity, we assume that each semantic rank has the same number of sentence pairs $n$ and represents model parameters with $\Theta$. In EMMA-X, we model the semantic relation of sentence pair $(\mathbf{x}_i, \mathbf{y}_k)$ through a joint distribution $P(\mathbf{x}_i, \mathbf{y}_k)$ with the semantic rank $c$ as a latent variable. Let $Q(c)$ be a prior distribution over the possible values of semantic ranks. That is $\sum_r Q(c = r) = 1, Q(c) \geq 0$. The training goal is to maximize the following likelihood:

$$
\begin{aligned}
\operatorname*{argmax}_{\Theta} \sum_{\mathbf{x}_i \in \mathcal{X}} \sum_{\mathbf{y}_k \in \mathcal{Y}} \log P(\mathbf{x}_i, \mathbf{y}_k | \Theta) &= \operatorname*{argmax}_{\Theta} \sum_{\mathbf{x}_i \in \mathcal{X}} \sum_{\mathbf{y}_k \in \mathcal{Y}} \log \sum_{r=1}^{N} P(\mathbf{x}_i, \mathbf{y}_k, c = r | \Theta) \\
&\geq \operatorname*{argmax}_{\Theta} \sum_{\mathbf{x}_i \in \mathcal{X}} \sum_{\mathbf{y}_k \in \mathcal{Y}} \sum_{r=1}^{N} Q(c = r) \log \frac{P(\mathbf{x}_i, \mathbf{y}_k, c = r | \Theta)}{Q(c = r)}.
\end{aligned}
\tag{12}
$$

**E-Step.** To make the inequality hold with equality, we have:

$$
Q(c = r) = \frac{P(\mathbf{x}_i, \mathbf{y}_k, c = r | \Theta)}{\sum_{j=1}^{N} P(\mathbf{x}_i, \mathbf{y}_k, c = j | \Theta)} = P(c = r | \mathbf{x}_i, \mathbf{y}_k, \Theta),
\tag{13}
$$

which is the posterior probability and is approximated by the prediction from GMM classifier. Since each sentence pair $(\mathbf{x}_i, \mathbf{y}_k)$ belongs to only one semantic rank, we approximate $Q(c = r) = \mathbb{I}(c_{\mathcal{G}}^* = r)$, which is a one-hot distribution.

**M-Step.** We try to maximize the likelihood in Eq. 12 under the semantic rank $c_{\mathcal{G}}^*$:

$$
\begin{aligned}
\operatorname*{argmax}_{\Theta} \sum_{\mathbf{x}_i \in \mathcal{X}} \sum_{\mathbf{y}_k \in \mathcal{Y}} \sum_{r=1}^{N} Q(r) \log \frac{P(\mathbf{x}_i, \mathbf{y}_k, r | \Theta)}{Q(r)} &\approx \operatorname*{argmax}_{\Theta} \sum_{\mathbf{x}_i \in \mathcal{X}} \sum_{\mathbf{y}_k \in \mathcal{Y}} \sum_{r=1}^{N} \log P(\mathbf{x}_i, \mathbf{y}_k | c_{\mathcal{G}}^* = r, \Theta) \\
&\geq \operatorname*{argmax}_{\Theta} n(n-1) \sum_{r=1}^{N} \tilde{\mu}_r^2,
\end{aligned}
\tag{14}
$$

The above derivation uses the assumption that $P(\mathbf{x}_i, \mathbf{y}_k | c_{\mathcal{G}}^* = r, \Theta) \sim \mathcal{N}_r(\mathbf{x}_i - \mathbf{y}_k | \tilde{\mu}_r, \tilde{\sigma}_r)$, with $\tilde{\mu}_r$ and $\tilde{\sigma}_r$ being the mean value and standard deviation of the Euclidean distance between sentence pairs in semantic rank $r$. Detailed proof of Eq. 14 is in **Appendix G**.

Next, we prove that minimizing the positive terms in expectation $\mathcal{L}_{\mathbf{CTL}}(\mathcal{X}, \mathcal{Y}; \Theta_{\mathcal{M}})$ actually equal to maximizing a lower bound of Eq. 14. As we apply dual supervision, data in the contrastive label space also follows the distribution $\mathcal{N}_r(\mathbf{x}_i - \mathbf{y}_k | \tilde{\mu}_r, \tilde{\sigma}_r)$. Hence, under mild assumptions, we can get:

$$
\mathcal{L}_{\mathbf{CTL}}^{+}(\mathcal{X}, \mathcal{Y}; \Theta_{\mathcal{M}}) = n^2 \sum_{r=1}^{N-1} \tilde{\mu}_r^2 < n(n-1) \sum_{r=1}^{N} \tilde{\mu}_r^2 \leq \sum_{\mathbf{x}_i \in \mathcal{X}} \sum_{\mathbf{y}_k \in \mathcal{Y}} \log P(\mathbf{x}_i, \mathbf{y}_k | \Theta),
\tag{15}
$$

where $\mathcal{L}_{\mathbf{CTL}}^{+}(\cdot)$ means the positive terms. In the derivation, we use the intrinsic property of semantic ranks ($\tilde{\mu}_1 < \tilde{\mu}_2 < ... < \tilde{\mu}_N$). Detailed proof is in **Appendix G**. Eq. 15 demonstrates that with dual supervision, minimizing the contrastive loss can partially maximize the likelihood in Eq. 12.

## 5 Experiments

To thoroughly evaluate the performance of EMMA-X, we conduct experiments on **XRETE** benchmark to verify the transfer ability of EMMA-X on various cross-lingual downstream tasks with strong baselines (pre-trained models: XLM-R [Conneau et al., 2020], INFOXLM [Chi et al., 2021], HICTL [Wei et al., 2021], sentence models: LaBSE [Feng et al., 2022], S-BERT [Reimers and Gurevych, 2020]) and ChatGPT in Section 5.2. See **Appendices C** and **D** for details. We further conduct geometric analysis in Section 5.3 to better interpret the cross-lingual transferability in EMMA-X.

### 5.1 Setup

**Corpus & Model.** We collect parallel corpora from CCAligned [El-Kishky et al., 2020], CCMatrix [Schwenk et al., 2021], WMT [Akhbardeh et al., 2021], and MultiUN [Ziemski et al., 2016], involving 94 languages with 3.2 billion sentence pairs. In addition, we add CC-100 [Conneau et al., 2020]

| Model | Inference | | Similarity | | Retrieval | | | | | Classification | | |
|---|---|---|---|---|---|---|---|---|---|---|---|---|
| | XNLI | ANLI | MultiSTS | QE | LAReQA | Mewsli-X | BUCC | Tatoeba | XCOPA | MultiEURLEX | MultiARC | PAWS-X |
| Metrics | Acc. (↑) | Acc. (↑) | Spearman (↑) | Pearson (↑) | mAP@20 (↑) | mAP@20 (↑) | F1 (↑) | Acc. (↑) | Acc. (↑) | Acc. (↑) | MAE (↓) | Acc. (↑) |
| MBERT* | 75.1[a] | - | 55.8[s] | - | 21.6[d] | 38.6[d] | 56.7[a] | 39.0[a] | 56.1[d] | 67.4[s] | 48.2[s] | 81.9[a] |
| XLM* | 77.8[b] | - | - | - | - | - | 56.8[a] | 32.6[a] | - | - | - | 80.9[a] |
| XLM-R* | 83.6[b] | 49.12[s] | 61.5[s] | 58.7[s] | 40.7[d] | 45.7[d] | 66.0[a] | 57.7[a] | 69.2[d] | 66.6[s] | - | 88.9[a] |
| HICTL* | 84.7[c] | - | - | - | - | - | 77.6[c] | 69.1[c] | - | - | - | 92.8[c] |
| CHATGPT† | 60.9 | 41.7 | 68.6 | 60.9 | - | - | - | - | 74.2 | 68.7 | 40.2 | 64.2 |
| *Ours re-implementation, translate-train-all (models are trained on English training data and on its data translated to the target language)* | | | | | | | | | | | | |
| XLM-R‡ | 82.8 | 48.48 | 65.9 | 63.2 | 40.3 | 48.6 | 67.9 | 59.1 | 71.2 | 66.9 | 44.9 | 90.1 |
| INFOXLM‡ | 84.2 | 49.10 | 82.2 | 64.1 | 44.9 | 57.1 | 77.4 | 66.2 | 74.6 | 67.7 | 36.2 | 93.0 |
| HICTL‡ | 85.1 | 49.02 | 81.6 | 64.9 | 46.1 | 54.8 | 77.6 | 65.8 | 74.8 | 68.3 | 38.2 | 92.8 |
| **EMMA-X** | **88.1** | **50.21** | **87.3** | **67.2** | **50.6** | **59.6** | **87.1** | **82.5** | **78.2** | **71.4** | **32.7** | **94.2** |

Table 1: Results on the XRETE benchmark. * denotes the results from previous literature, [a]Hu et al. [2020] [b]Conneau et al. [2020] [c]Wei et al. [2021] [d]Ruder et al. [2021]. [s] denotes the results from the original paper. ‡ denotes results from our re-trained models with the same model size and training corpora as EMMA-X. † denotes the zero-shot performance. Refer to **Appendix F** for greater details on each task and language.

| Model | Similarity | | Retrieval | | | |
|---|---|---|---|---|---|---|
| | MultiSTS | QE | LAReQA | Mewsli-X | BUCC | Tatoeba |
| Metrics | Spearman (↑) | Pearson (↑) | mAP@20 (↑) | mAP@20 (↑) | F1 (↑) | Acc. (↑) |
| S-BERT | **84.0** | 39.3 | **31.8** | **14.4** | **88.5** | 68.6 |
| LaBSE | **74.4** | 31.6 | 12.8 | 11.2 | **93.2** | **83.6** |
| INFOXLM | 55.2 | **49.4** | 16.9 | 23.9 | 77.4 | 66.2 |
| XLM-R | 25.0 | 10.4 | 16.1 | 11.7 | 67.9 | 59.1 |
| **EMMA-X** | 62.9 | **54.7** | 19.4 | 29.4 | 87.1 | 82.5 |

Table 2: Zero-shot Results on Similarity and Retrieved tasks. Results of LaBSE use Customized Vocab setting. Results of S-BERT are from **XLM-r←SBERT-paraphrases**. The bold font denotes the best 2 results.

| Model | FLORES-200 | | Tatoeba | |
|---|---|---|---|---|
| | Head | Long-tail | Head | Long-tail |
| Metrics | Acc. (↑) | Acc. (↑) | Acc. (↑) | Acc. (↑) |
| S-BERT | 87.5 | 51.6 | 91.4 | 57.8 |
| LaBSE | 99.9 | 82.5 | **95.7** | 77.9 |
| INFOXLM | 83.4 | 53.8 | 87.8 | 56.1 |
| XLM-R | 68.2 | 45.3 | 66.3 | 55.7 |
| **EMMA-X** | 94.5 | **84.2** | 91.9 | **78.1** |

Table 3: Retrieval results on FLORES-200 and Tatoeba in xx → En direction. The bold font denotes the best results.

as the large-scale monolingual corpus with about 800 billion sentences that covers 94 languages. The cross-lingual encoder starts from the well-trained XLM-R large model [Conneau et al., 2020]. The GMM classifier is implemented as a mixture of Gaussian forms, each of which consists of a prior $\pi \in \mathbb{R}^1$, a mean $\mu \in \mathbb{R}^{1024}$ and a standard deviation $\sigma \in \mathbb{R}^{1024}$, all are trainable variables. We set the total semantic ranks as $N = 4$. The statistics of all data and hyper-parameters are shown in **Appendix A**.

## 5.2 XRETE Evaluation

**XRETE** includes 12 cross-lingual tasks divided into 4 different categories. We report the "translate-train" performance in Table 1 on most tasks but zero-shot performance on BUCC and Tatoeba following [Ruder et al., 2021]. Table 2 presents zero-shot comparisons with sentence models.

**Comparisons with Pre-trained Models.** In Table 1, EMMA-X consistently outperforms all baseline models (XLM-R [Conneau et al., 2020], HICTL [Wei et al., 2021] and INFOXLM [Chi et al., 2021]) with 7.97% improvements on average. Specifically, EMMA-X achieves 88.1% accuracy on XNLI [Conneau et al., 2018] and 50.21% accuracy on ANLI [Ebrahimi et al., 2022] with up to 6.4% improvements than baselines. On the MultiSTS [Reimers and Gurevych, 2020] task, EMMA-X achieves an 87.3 correlation score, outperforming several strong baselines by 5.1∼21.4, and even achieves comparable performance in the cross-lingual and the monolingual settings (see **Appendix F** for language-specific results). Furthermore, EMMA-X obtains a 67.2 Pearson score on QE [Specia et al., 2021] task, which is comparable to the winner on the leaderboard[3] without any specific finetuning techniques. As for sentence retrieval, EMMA-X consistently outperforms previous strong baselines among all 4 tasks [Ruder et al., 2021], and demonstrates 2.6%∼39.6% improvements over these baselines. Similar results can be found in sentence classification tasks. EMMA-X obtains an 81.3% top-1 accuracy averaged on XCOPA [Ponti et al., 2020], MultiEURLEX [Chalkidis et al., 2021] and PAWS-X [Yang et al., 2019b] tasks, outperforming XLM-R, INFOXLM and HICTL by 7.0%, 3.9% and 3.5% improvements, respectively. On MultiARC [Keung et al., 2020] task, EMMA-X shows the lowest error rates among all models. The consistent improvements on all tasks reveal that EMMA-X can obtain better universal representations for different natural languages with various

---

[3]`https://www.statmt.org/wmt21/quality-estimation-task_results.html`

topics and domains. We further conduct experiments with ChatGPT on XRETE tasks without 4 Retrieval tasks. We list the prompts for each task in **Appendix E**. ChatGPT's zero-shot performance is worse than fine-tuned pre-trained models and the performance gap is very large on most tasks.

**Zero-shot comparisons with sentence models.** Compared with XLM-R and INFOXLM, which adopt the same amount of training data as EMMA-X, EMMA-X consistently outperforms XLM-R and INFOXLM by 73.2% and 25.1% on average, as shown in Table 2. The results further prove the effectiveness of our pre-training technique. Through the reciprocation between GMM classifier and cross-lingual encoder, EMMA-X can generate reliable semantic rank for multilingual non-parallel corpora, which can provide more supervision signals than previous baselines. EMMA-X even achieves comparable results with strong supervised methods: LaBSE [Feng et al., 2022] and S-BERT [Reimers and Gurevych, 2020], which both trained on supervised data. LaBSE is trained on a fine-filtered bilingual corpus with 6B translation pairs (2 times larger than EMMA-X), while S-BERT is distilled from a S-BERT model fine-tuned on English NLI, STS datasets, and 50M English paraphrase pairs. Compared with these two methods, EMMA-X can achieve the best results on QE and Mewsli-X by outperforming S-BERT and LaBSE by 71.7% and 117.8% averaged. EMMA-X performs worse than these baselines on MultiSTS and BUCC, for these two tasks only contain rich-resource languages, which already have great deal of parallel data.

| Model | Head Langs. | | | Long-tail Langs. | | | All Langs. | | |
|---|---|---|---|---|---|---|---|---|---|
| | Invariance | Canonical Form | Isotropy | Invariance | Canonical Form | Isotropy | Invariance | Canonical Form | Isotropy |
| Metrics | KL-D ($\downarrow$) | CH-I ($\uparrow$) | PR ($\uparrow$) | KL-D ($\downarrow$) | CH-I ($\uparrow$) | PR ($\uparrow$) | KL-D ($\downarrow$) | CH-I ($\uparrow$) | PR ($\uparrow$) |
| XLM-R (cls) | 0.7356 | 30.19 | 0.3681 | 2.0042 | 7.96 | 0.3686 | 1.6501 | 20.80 | 0.3683 |
| INFOXLM (cls) | 0.4491 | 38.82 | 0.4478 | 1.8555 | 13.02 | 0.4406 | 1.4747 | 31.51 | 0.4665 |
| S-BERT (mean) | **0.1115** | **108.22** | 0.4519 | 1.3112 | 44.32 | 0.4414 | **0.9782** | **102.36** | 0.4467 |
| EMMA-X (cls) | 0.3603 | 43.52 | **0.5318** | **0.3963** | **46.53** | **0.5732** | 1.1904 | 48.70 | **0.5918** |

Table 4: Comparisons with existing methods on FLORES dataset for geometric analysis. "cls" and "mean" represent different pooling strategies to obtain sentence representations.

**Performance on Long-tail Languages.** One goal of EMMA-X is to learn universal sentence representations accommodated for more languages. To better prove this, we report the retrieval accuracy on FLORES-200 [Costa-jussà et al., 2022] and Tatoeba [Artetxe and Schwenk, 2019]. We reformulate FLORES-200, which contains manual translations in 204 languages (totaling 3001 sentences) to perform retrieval tasks in the same way as Tatoeba and report the performance in terms of language data scale in Table 3. Details about the separation of languages and FLORES-200 are shown in **Appendices A** and **B**. On head languages, EMMA-X performs worse than LaBSE by about 4.6% but outperforms S-BERT by 3.5%. On the long-tail languages, EMMA-X can surpass S-BERT by 4.3% averaged on two tasks. EMMA-X can even exceed the strongest LaBSE by 2.1% on FLORES. One reason for the superior results on long-tail languages is that for those long-tail languages that have only bi-lingual parallel data with rich-resource languages (often English), EMMA-X can provide multi-lingual semantic relation signals for them with arbitrary languages through dual supervision.

### 5.3 Geometric Analysis

To interpret the advantages of EMMA-X, we evaluate the geometric characteristics of it on FLORES-200 dataset [Costa-jussà et al., 2022] without any fine-tuning. The criteria of three requirements are Invariance, measured with KL-divergence (KL-D) [Kullback and Leibler, 1951], Canonical Form, measured with Calinski-Harabasz Index (CH-I) [Caliński and Harabasz, 1974] and Isotropy, measured with principal ratio (PR) [Mu and Viswanath, 2018]. Details of them are shown in **Appendix B**. We report the numerical results in Table 4 and visualize each characteristic in Figure 2.

**Invariance & Canonical Form** aim to measure how languages are aligned in the representation space. If the sentence representations are universal, then sentences in different languages should follow a similar distribution, which is measured by invariance in KL-divergence. Similarly, canonical form measures how well semantic equivalent sentences are grouped into one cluster, with a clustering metric (CH-I). In Table 4, S-BERT outperforms other baselines in "Invariance" and "Canonical Form" on head languages. However, EMMA-X shows better performance on long-tail languages in these two metrics, which is consistent with Table 3. Figure 2 presents similar results. Among the 20 languages we randomly sampled from FLORES, EMMA-X can align 17 languages as shown in Figure 2d, with "xh, eo, ur" as outliers. In Figure 2e, 2g, 2f and 2h, different colors represent different languages. So

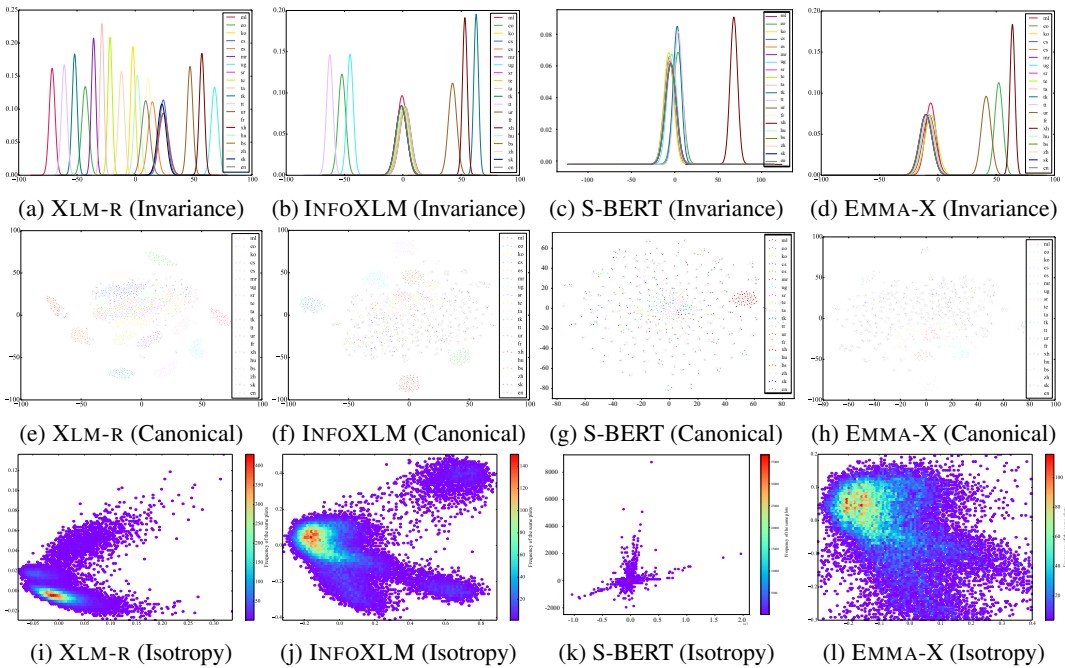

| (a) XLM-R (Invariance) | (b) INFOXLM (Invariance) | (c) S-BERT (Invariance) | (d) EMMA-X (Invariance) |
| (e) XLM-R (Canonical) | (f) INFOXLM (Canonical) | (g) S-BERT (Canonical) | (h) EMMA-X (Canonical) |
| (i) XLM-R (Isotropy) | (j) INFOXLM (Isotropy) | (k) S-BERT (Isotropy) | (l) EMMA-X (Isotropy) |

Figure 2: Visualization of representations from EMMA-X, XLM-R, INFOXLM and S-BERT. We use t-SNE [van der Maaten and Hinton, 2008] to visualize each geometric metrics.

a cluster with only one color means this language is isolated from other languages and not aligned well in representation space. Figure 2h shows that EMMA-X performs well in most languages.

**Isotropy** measures how expressive a representation space is since high-dimensional representation space can easily deteriorate into a low-dimensional manifold. From Table 4, EMMA-X achieves the best results in isotropy. The Isotropy of S-BERT (k) is different from other methods. We conjecture the reason is that S-BERT removes MLM tasks during fine-tuning, so token embeddings will only receive sentence-level supervision, resulting in identical sentence representations for different languages. The abnormal results observed on the high KL-divergence, as depicted in Table 4 and Figure 2c, can be attributed to the representation space for S-BERT deteriorating into a low-dimensional manifold (low isotropy score in Table 4 and Figure 2k), and different languages are not distributed uniformly across the whole representation space, which limits the expressive ability.

| Model\Language | af | ar | bg | bn | de | el | es | et | eu | fa | fi | fr | he | hi | hu | id | it | ja |
|---|---|---|---|---|---|---|---|---|---|---|---|---|---|---|---|---|---|---|
| + Phase 1 | 86.7 | 84.9 | 87.8 | 80.0 | **96.6** | 89.8 | 91.9 | 90.3 | 81.5 | 84.3 | 88.2 | 88.3 | 85.2 | 84.9 | 89.1 | **93.7** | 88.7 | 86.2 |
| + fixed GMM | 86.2 | 84.3 | 89.6 | 79.4 | 92.6 | 87.5 | 93.9 | 92.6 | 84.5 | 87.9 | 92.1 | 91.4 | 84.4 | 92.6 | 87.1 | 90.9 | 87.1 | 82.4 |
| EMMA-X | **93.9** | **90.2** | **91.4** | **92.1** | 95.9 | **92.0** | **95.7** | **94.3** | **96.4** | **96.0** | **94.7** | **92.1** | **90.3** | **98.5** | **92.0** | 93.4 | **91.6** | **91.3** |
| Model\Language | jv | ka | kk | ko | ml | mr | nl | pt | ru | sw | ta | te | th | tl | tr | ur | vi | zh |
| + Phase 1 | 22.6 | 88.3 | 61.6 | 85.0 | 93.9 | 82.0 | 94.7 | 92.1 | 88.7 | 70.5 | 70.9 | 81.5 | 84.9 | 59.2 | 88.2 | 85.8 | 97.1 | 91.7 |
| + fixed GMM | 37.9 | 91.8 | 67.1 | 87.6 | 91.8 | 85.3 | 93.1 | 88.9 | 90.8 | 75.4 | 72.4 | 82.3 | 86.7 | 68.1 | 91.6 | 89.3 | 94.9 | 88.8 |
| EMMA-X | **56.7** | **93.2** | **86.0** | **91.4** | **96.7** | **94.8** | **93.6** | **92.7** | **91.6** | **87.2** | **87.8** | **95.4** | **96.9** | **85.7** | **96.5** | **93.4** | **96.1** | **92.7** |

Table 5: Ablation Study about EMMA-X on 36 languages in Tatoeba. We mark the 14 long-tail languages with color red. We highlight the best results with bold font and the second-best results with underlines.

## 5.4 Ablation Analysis

The primary goal of EMMA-X is to acquire universal semantic representations for a multitude of languages. For the limited number of parallel data, EMMA-X strives to leverage non-parallel data to extend language coverage for universal semantic representations and to enhance representation performance for those languages. To provide additional evidence supporting our claim, we propose an ablation experiment in Table 5. The results in Table 5 demonstrate that EMMA-X significantly

| Setting | XNLI Acc. (↑) | ANLI Acc. (↑) | MultiSTS Spearman (↑) | QE Pearson (↑) | LAReQA mAP@20 (↑) | Mewsli-X mAP@20 (↑) | BUCC F1 (↑) | Tatoeba Acc. (↑) | XCOPA Acc. (↑) | MultiEURLEX Acc. (↑) | MultiARC MAE (↓) | PAWS-X Acc. (↑) |
|---|---|---|---|---|---|---|---|---|---|---|---|---|
| (FCN, $N=4$) | 86.4 | 46.63 | 87.0 | 66.1 | 46.6 | 54.2 | 81.1 | 78.2 | 76.2 | 68.5 | 34.9 | 90.8 |
| (GMM, $N=2$) | 85.2 | 49.52 | 84.7 | 67.3 | 47.8 | 56.6 | 85.6 | 82.3 | 77.0 | 70.1 | **30.6** | 90.9 |
| (GMM, $N=4$)‡ | **88.1** | 50.21 | **87.3** | 67.2 | **50.6** | **59.6** | **87.1** | 82.5 | 77.4 | **71.4** | 32.7 | 94.2 |
| (GMM, $N=8$) | 87.9 | **50.58** | 86.3 | **67.4** | 49.2 | 59.5 | 85.7 | 82.5 | **78.8** | 71.3 | 32.5 | **95.1** |

Table 6: Effect of different settings on the XRETE benchmark. ‡ means results with default settings (i.e., GMM classifier and $N = 4$). "FCN" means the semantic classifier is developed as fully connected networks.

improves the retrieval accuracy by 8.1 when compared to solely using parallel data. Moreover, on 16 long-tail languages, the retrieval accuracy further increases from 76.2 to 90.7, representing a substantial improvement of 19.0%. These findings support our claim that EMMA-X can indeed learn effective universal semantic representations from an abundance of non-parallel data.

### 5.5 Semantic Rank and GMM Analysis

We assessed EMMA-X's performance on XRETE benchmark with various settings in Table 6. The evaluation focuses on two aspects: the classifier type and the value of semantic similarity rank.

**Type of the classifier.** Without direct supervision, we propose each semantic rank follow a Gaussian distribution. Comparing "**GMM**" with "**FCN**" in Table 6, the performance drops from 70.7 to 68.1. This decline is attributed to the handling of outliers: the "**GMM**" handles them better due to its soft assignment approach, giving outliers low probabilities and thereby minimizing their influence.

**Value of Semantic Similarity Rank.** Different semantic similarity ranks were evaluated in Table 6. The peak performance at $N = 4$ implies that a refined similarity rank ensures a balanced distribution of semantic ranks, which helps alleviate data imbalance. However, increasing the number of semantic ranks complicates the learning process for both "**GMM**" and the cross-lingual encoder. $N = 4$ strikes a balance between data distribution and learning complexity.

## 6 Related Work

**Cross-lingual Representation Pre-training** In recent years, various studies [Devlin et al., 2019, Conneau et al., 2020] have shifted the monolingual pre-training procedure to multilingual scenarios. Most of them often rely heavily on parallel data to learn cross-lingual sentence representations, with several improved techniques, such as language modeling [Conneau and Lample, 2019] and contrastive learning [Chi et al., 2021, Hu et al., 2021, Wei et al., 2021]. Recently, some endeavors incorporate monolingual data into parallel corpora, by translating monolingual corpora into pseudo parallel corpora [Kvapilíková et al., 2020, Ouyang et al., 2021], or computing the semantic similarity in monolingual corpora off-the-shelf and using it as supervision signals [Goswami et al., 2021].

**Contrastive Learning** has become a popular paradigm in NLP. Besides constructing the positives and negatives through parallel corpora [Zhang et al., 2021] or other labeled data [Gunel et al., 2021, Ni et al., 2022], researchers also adopt self-supervised methods, which build positives and negatives by corrupting sentences [Gao et al., 2021, Chuang et al., 2022] or data augmentation methods [Zhang et al., 2022a, Wu et al., 2022]. Another line of research improves the quality of negatives by preserving a memory queue [Yang et al., 2021, Wang et al., 2021] or generating high-quality negatives [Zhang et al., 2022c, Wei et al., 2022].

## 7 Conclusion

In this paper, we study the problem of learning cross-lingual universal representations. Major contributions are fourfold: 1) We propose a novel paradigm EMMA-X, which can make full use of massive monolingual data to learn universities for any two languages. 2) We summarize three requirements for the universal representation space among all languages and verify the superiority of EMMA-X towards strong baselines. 3) To incentivize the research of cross-lingual universal representation learning, we form a novel benchmark (XRETE) with 12 cross-lingual tasks fully depending on sentence-level representations. 4) Experiments on XRETE demonstrate that EMMA-X achieved state-of-the-art results over strong baselines.

## Acknowledgements

We thank the anonymous reviewers for their constructive comments. This work is supported by the National Natural Science Foundation of China (Grant No.U21B2009). This research is also supported by the Strategic Priority Research Program of the Chinese Academy of Science, Grant No.XDC02030400.

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

# A   Training Corpora and Hyper-parameters

## A.1   Training Corpora

As for monolingual data, we follow Conneau et al. [2020] to build a Common-Crawl Corpus using the CCNet [Wenzek et al., 2020] tool[4], which is widely used in the literature Huang et al. [2019], Luo et al. [2021], Chi et al. [2021], Wei et al. [2021]. Further, we collect parallel corpora from CCAligned El-Kishky et al. [2020], CCMatrix Schwenk et al. [2021], WMT Akhbardeh et al. [2021], and MultiUN Ziemski et al. [2016], involving 94 languages with more than 4.8 billion sentence pairs. We use the OpusFilter[5] tool to remove noisy bitexts, which results in 3.2 billion sentence pairs. Table 7 shows the statistics for both monolingual and parallel data. We apply subword tokenization directly on raw text data using Sentence Piece Model Kudo and Richardson [2018] without any additional preprocessing. To better support our motivation that EMMA-X can cover more languages than previous cross-lingual sentence representations, we divide Tatoeba Artetxe and Schwenk [2019] into two subsets: "Head", containing languages usually covered in previous methods, and "Long-tail", with other languages. We treat the 36 languages contained in XTREME Ruder et al. [2021] as head languages, which are: **"ar, he, vi, id, jv, tl, eu, ml, ta, te, af, nl, en, de, el, bn, hi, mr, ur, fa, fr, it, pt, es, bg, ru, ja, ka, ko, th, sw, zh, kk, tr, et, fi, hu, az, lt, pl, uk, ro"**. The remaining 76 languages in Tatoeba are treated as long-tail ones.

## A.2   Hyper-parameters

The parameters of EMMA-X are first initialized with XLM-R, with 24 layers of Transformer [Vaswani et al., 2017] encoder, 1024 hidden states, and 16 attention heads. We set the total semantic ranks as 4. The GMM classifier is implemented as a mixture of Gaussian forms, each of which consists of a prior $\pi \in \mathbb{R}^1$, a mean $\mu \in \mathbb{R}^{1024}$, and a variance $\sigma \in \mathbb{R}^{1024}$, all are trainable variables. We optimize the GMM classifier with Adam ($\beta_1$=0.9, $\beta_2$=0.999) Kingma and Ba [2015] using a batch size of 1024 and a learning rate of 3e-5. For cross-lingual encoder, we apply the same training setting as MoCo He et al. [2020], with the momentum queue $K$ to be 256 and temperature as 0.04. We set the momentum coefficient to 0.999 and use the Adam optimizer with a cosine decay learning rate whose peak is 5e-4.

# B   FLORES-200 Dataset and Geometric Analysis

## B.1   FLORES-200 dataset

FLORES-200 Goyal et al. [2022], Costa-jussà et al. [2022] is a many-to-many multilingual benchmark, which consists of 3001 sentences in 204 total languages. FLORES-200 sourced all sentences from English WikiMedia and translated these English sentences into 204 languages by human translators. In particular, sentences in FLORES-200 have a much larger breadth of topics, for they are collected from three different sources: WikiNews[6], WikiJunior[7] and WikiVoyage[8]. We summarize the basic statistics of all languages in FLORES-200 in Table 8. Similar to Tatoeba [Artetxe and Schwenk, 2019], we treat English data "eng_Latn" as retrieval labels and report the retrieval accuracy using the same scripts as Tatoeba in XTREME [Ruder et al., 2021]. We set the 68 languages: **"bel_Cyrl, bos_Latn, hun_Latn, epo_Latn, khm_Khmr, urd_Arab, srp_Cyrl, jav_Latn, hye_Armn, gla_Latn, por_Latn, lit_Latn, bul_Cyrl, slk_Latn, mal_Mlym, ita_Latn, nno_Latn, mar_Deva, hrv_Latn, hin_Deva, kat_Geor, ben_Beng, fin_Latn, cym_Latn, oci_Latn, cat_Latn, fao_Latn, xho_Latn, spa_Latn, ron_Latn, amh_Ethi, ces_Latn, swe_Latn, nld_Latn, tat_Cyrl, kor_Hang, glg_Latn, fra_Latn, eus_Latn, ind_Latn, dan_Latn, tha_Thai, deu_Latn, tel_Telu, afr_Latn, pol_Latn, est_Latn, uig_Arab, ukr_Cyrl, uzn_Latn, heb_Hebr, kaz_Cyrl, nob_Latn, rus_Cyrl, vie_Latn, arb_Arab, zho_Hans, tuk_Latn, khk_Cyrl, jpn_Jpan, ell_Grek, isl_Latn, tam_Taml, slv_Latn, tur_Latn, mkd_Cyrl, tgl_Latn, gle_Latn"** as "Head" languages, and the remaining 135 languages (excluded English data) as "Long-tail" ones.

---

[4]https://github.com/facebookresearch/cc_net
[5]https://github.com/Helsinki-NLP/OpusFilter
[6]https://en.wikinews.org/wiki/MainPage
[7]https://en.wikibooks.org/wiki/Wikijunior
[8]https://en.wikivoyage.org/wiki/Main_Page

| Code | Size (GB) | Sent. (M) | Code | Size (GB) | Sent. (M) | Code | Size (GB) | Sent. (M) | Code | Size (GB) | Sent. (M) | Code | Size (GB) | Sent. (M) |
|---|---|---|---|---|---|---|---|---|---|---|---|---|---|---|
| af | 1.3 | - | et | 6.1 | 22.3 | ja | 24.2 | 89.2 | mt | 0.2 | - | sq | 3.0 | - |
| am | 0.7 | - | eu | 2.0 | 0.81 | jv | 0.2 | - | my | 0.9 | - | sr | 5.1 | - |
| ar | 20.4 | 72.3 | fa | 21.6 | 7.5 | ka | 3.4 | 2.0 | ne | 2.6 | - | su | 0.1 | - |
| as | 0.1 | - | fi | 19.2 | 92.8 | kk | 2.6 | 2.8 | nl | 15.8 | 66.0 | sv | 10.8 | 74.2 |
| az | 3.6 | 0.82 | fr | 46.5 | 331.5 | km | 1.0 | 0.84 | no | 3.7 | - | sw | 1.6 | 1.7 |
| be | 3.5 | 0.51 | fy | 0.2 | 0.13 | kn | 1.2 | - | om | 0.1 | - | ta | 8.2 | 2.79 |
| bg | 22.6 | 47.2 | ga | 0.5 | - | ko | 17.2 | 79.3 | or | 0.6 | - | te | 2.6 | - |
| bn | 7.9 | 7.52 | gd | 0.1 | 0.05 | ku | 0.4 | - | pa | 0.8 | - | th | 14.7 | 13.1 |
| br | 0.1 | - | gl | 2.9 | 0.77 | ky | 1.2 | - | pl | 16.8 | 79.7 | tl | 0.8 | - |
| bs | 0.1 | - | gu | 0.3 | - | la | 2.5 | - | ps | 0.7 | - | tr | 17.3 | 93.8 |
| ca | 10.1 | 14.9 | ha | 0.3 | - | lo | 0.6 | - | pt | 15.9 | 247.6 | ug | 0.4 | - |
| cs | 16.3 | 108.4 | he | 6.7 | 47.1 | lt | 7.2 | 11.0 | ro | 8.6 | 60.4 | uk | 9.1 | 0.78 |
| cy | 0.8 | - | hi | 20.2 | 3.2 | lv | 6.4 | 0.37 | ru | 48.1 | 134.9 | ur | 5.0 | 1.15 |
| da | 15.2 | 8.0 | hr | 5.4 | - | mg | 0.2 | - | sa | 0.3 | - | uz | 0.7 | - |
| de | 46.3 | 283.4 | hu | 9.5 | 55.2 | mk | 1.9 | - | sd | 0.4 | - | vi | 44.6 | 15.3 |
| el | 29.3 | 95.1 | hy | 5.5 | 1.7 | ml | 4.3 | 1.07 | si | 2.1 | 0.60 | xh | 0.1 | - |
| en | 49.7 | - | id | 10.6 | 184.6 | mn | 1.7 | 0.19 | sk | 4.9 | - | yi | 0.3 | - |
| eo | 0.9 | 0.18 | is | 1.3 | - | mr | 1.3 | - | sl | 2.8 | 9.8 | zh | 36.8 | 379.4 |
| es | 44.6 | 279.6 | it | 19.8 | 179.3 | ms | 3.2 | 2.1 | so | 0.4 | - | - | | - |

Table 7: The statistics of CC-100 and the collected parallel corpora used for training. We report the list of 94 languages and include the size of the monolingual data (in GiB) and the number of sentence pairs (in Millions, which denotes the number of sentence pairs between the specific language and English) in parallel corpora for each language. "-" means the number of sentence pairs is less than 0.1 million.

| | |
|---|---|
| Number of Sentences | 3001 |
| Average Words per Sentence | 21 |
| Number of Articles | 842 |
| Average Number of Sentences per Article | 3.5 |

| Domain | Articles | Sentences |
|---|---|---|
| WikiNews | 309 | 993 |
| WikiJunior | 284 | 1006 |
| WikiVoyage | 249 | 1002 |

| Sub-Topic | Articles | Sentences |
|---|---|---|
| Crime | 155 | 313 |
| Disasters | 27 | 65 |
| Entertainment | 28 | 68 |
| Geography | 36 | 86 |
| Health | 27 | 67 |
| Nature | 17 | 45 |
| Politics | 171 | 341 |
| Science | 154 | 325 |
| Sports | 154 | 162 |
| Travel | 505 | 1529 |

Table 8: Basic Statistics of FLORES-200.

## B.2 Three measurements in Geometric Analysis

**Invariance Measurement** implies whether the semantic distributions of all languages are similar [Abend and Rappoport, 2017]. We adopt a Gaussian form $\mathcal{N}_l(\mu_l, \sigma_l^2)$ where $\mu_l = \frac{\sum_{\mathbf{x} \in l} \gamma^{(\mathbf{x})}}{3001}$ and $\sigma_l^2 = \sum_{\mathbf{x} \in l} (\gamma^{(\mathbf{x})} - \mu_l)(\gamma^{(\mathbf{x})} - \mu_l)^T$, to approximate the semantic distribution of each language $l$. Further, we compute the mean averaged KL-divergence (KL-D for short) [Kullback and Leibler, 1951] among all language pairs as the overall Invariance score $\mathcal{I}_v$ with $L$ as the total number of languages:

$$\mathcal{I}_v = \frac{1}{L \times (L-1)} \sum_{l_1 \neq l_2} \frac{\mathbf{KL}(\mathcal{N}_{l_1} || \mathcal{N}_{l_2}) + \mathbf{KL}(\mathcal{N}_{l_2} || \mathcal{N}_{l_1})}{2}. \tag{16}$$

**Canonical Form Measurement** Previous works [Teller, 2000, Irwin et al., 2009] have demonstrated that a good multilingual space should distribute sentence representations based on their semantic similarities rather than language families. To measure this in quantity, we focus on Calinski-Harabasz Index (CH-I) [Caliński and Harabasz, 1974], which measures how similar an object is to its own cluster compared to other clusters. We group all semantically equivalent sentences in a cluster, which leads to 3001 clusters and each obsesses 204 sentences in 204 different languages. Assuming $c_k$ and $c$ are the centroid of cluster $k$ and the whole dataset $s$, respectively. The CH-I $\mathcal{C}_h$ is defined as:

$$\mathcal{C}_h = \left[ 204 \times \sum_{k=1}^{K} \|c_k - c\|^2 \right] / \left[ \sum_{k=1}^{K} \sum_{s \in \mathcal{S}} \|s - c_k\|^2 \right]. \tag{17}$$

| Task category | Task | Train | Dev | Test | Lang. | Metric | Domain |
|---|---|---|---|---|---|---|---|
| Inference | AmericasNLI | 392,702 | 222-743 | 738-750 | 10 | Accuracy | Misc. |
| | XNLI | 392,702 | 2,490 | 5,010 | 15 | Accuracy | Misc. |
| Semantic Similarity | Multi-STS | 550,152+5,749 | 10,000+1,500 | 250 | 7 | Spearman | Misc. |
| | WMT21QETask1 | 7,000 | 1,000 | 1,000 | 7 (11) | Pearson | News |
| Sentence Retrieval | LAReQA | 87,599 | 10,579 | 1,190 | 11 | mAP@20 | Wikipedia |
| | Mewsli-X | 116,093 | 10,252 | 428-1,482 | 11 (50) | mAP@20 | News |
| | BUCC | - | - | 1,896-14,330 | 5 | F1 | Wiki/News |
| | Tatoeba | - | - | 1,000 | 36 (122) | Accuracy | Misc. |
| Classification | XCOPA | 33,410+400 | 100 | 500 | 11 | Accuracy | Misc. |
| | MultiEURLEX | 55,000 | 5,000 | 5,000 | 23 | Accuracy | Legal |
| | MultiARC | 200,000 | 5,000 | 5,000 | 6 | MAE | Reviews |
| | PAWS-X | 49,401 | 2,000 | 2,000 | 7 | Accuracy | Wiki/Quora |

Table 9: Overview of XRETE tasks. For tasks that have training and dev sets in other languages, we only report the number of sentences in English sets. We report the number of test examples per language.

The higher the CH-I is, the better the semantically equivalent sentences are clustered.

**Isotropy Measurement** A high-dimensional embedding space often demonstrates poor isotropy, and deteriorates into a low-dimensional manifold that greatly limits the expressive ability of the embedding space. We adopt principal ratio (PR) [Mu and Viswanath, 2018] to measure isotropy. Let $\varepsilon$ be the sentence representation matrix, $\mathcal{V}$ be the set of the eigenvectors of $\varepsilon$, the Isotropy $\mathcal{I}_{so}$ is

$$\mathcal{I}_{so} = \min_{v \in \mathcal{V}} \sum_{e \in \mathcal{E}} \exp(v^\mathsf{T} e) / \max_{v \in \mathcal{V}} \sum_{e \in \mathcal{E}} \exp(v^\mathsf{T} e). \tag{18}$$

The closer $\mathcal{I}_{so}$ is to 1, the more isotropic the representation space is.

## C    XRETE: Cross-lingual Representation Transfer Evaluation

XRETE consists of 12 tasks that fall into four different categories. In our "translate-train-all" setting, we individually fine-tune models with English training set and its translated training sets on each task. Then we report the performance of our fine-tuned model. We give an overview in Table 9 and describe the task details as follows.

**XNLI** The Cross-lingual Natural Language Inference corpus Conneau et al. [2018] tasks the systems with reading two sentences and determining whether one entails the other, contradicts it, or neither (neutral). A crowdsourcing-based procedure is used for collecting English examples, which are later translated into ten target languages for evaluation. Training data stays consistent with the English training data of MultiNLI Williams et al. [2018]. For evaluation, we concatenate two sentences as input and apply a new classification head to distinguish sentence relationships. We perform "translate-train-all" evaluation, where the model is first fine-tuned on English training data and its translated data in other languages, then evaluated on test sets.

**AmericasNLI (ANLI)** The AmericasNLI Ebrahimi et al. [2022] is an extension of XNLI task to 10 Indigenous languages of the Americas. All of these languages are truly low-resource languages and serve as a good testbed for zero-shot cross-lingual transferability. As Spanish is more relative to the target languages, the Spanish version of XNLI subset is translated for evaluation. For training, both English and Spanish versions of MultiNLI training data are provided. We evaluate on ANLI following the same settings as in XNLI.

**MultiSTS** The Multilingual Semantic Textual Similarity dataset Cer et al. [2017], Reimers and Gurevych [2020] aims to assign a semantic similarity score for a pair of sentences. The MultiSTS dataset contains 7 cross-lingual sentence pairs and 3 monolingual pairs. Stanford NLI Bowman et al. [2015] and English STS Cer et al. [2017] are provided as training sets. We report the results after first fine-tuning on English training set using a Siamese network structure [Reimers and Gurevych, 2020]. Then we compute the cosine similarity between the sentence pairs and compute Spearman's rank correlation between the predicted score and gold score following Reimers and Gurevych [2020].

**WMT21QETask1 (QE)**  The WMT21 Quality Estimation Task 1 Sentence-level Direct Assessment Specia et al. [2021] aims at testing the translation quality and this task has been applied to test the sensitivity of language models to semantic similarity Tiyajamorn et al. [2021]. The training and evaluation sets are collected from Wikipedia by translating sentences using state-of-the-art translation models to 6 languages and annotated by professional translators. In WMT21, 4 new language pairs with no training data are given to test zero-shot cross-lingual transferability. Our evaluation setting on QE is similar to that on MultiSTS, but we report Pearson's rank correlation [Kepler et al., 2019].

**LAReQA**  The Language-Agnostic Retrieval Question Answering Roy et al. [2020] is a QA retrieval task where models are required to retrieve all relevant answers in different languages over a large multilingual pool. The dataset is constructed on XQuAD Artetxe et al. [2020] and a question is linked with answer sentences in different languages. The training set of SQuAD v1.1 Rajpurkar et al. [2016] is used to fine-tune the model to adapt to QA retrieval task. During the evaluation, sentence embeddings are also obtained by a siamese network, and we retrieve the sentences with the highest cosine similarity as predictions.

**Mewsli-X**  Mewsli (**M**ultilingual **E**ntities in **N**ews, **li**nked) requires linking an entity mention to its entry in a language-agnostic knowledge base Botha et al. [2020]. Mewsli-X Ruder et al. [2021] features 15k mentions in 11 languages. For each mention, Mewsli-X offers entity descriptions candidate pool containing 1M candidates across 50 languages. Fine-tuning is done on a predefined set of English-only mention-entity pairs from English Wikipedia hyperlinks. Our evaluation setting is identical to LAReQA.

**BUCC**  The second and third shared task of the workshop on Building and Using Parallel Corpora Zweigenbaum et al. [2017], Pierre Zweigenbaum and Rapp [2018] aims to examine the ability of models to detect parallel sentence pairs in a pair of monolingual corpora. The dataset provides train and test splits in 5 languages. Following XTREME Hu et al. [2020], we directly evaluate on BUCC without fine-tuning and retrieve sentences with the highest cosine similarity.

**Tatoeba**  The goal of the Tatoeba dataset Artetxe and Schwenk [2019] is to find the nearest neighbor for each sentence in the other language according to cosine similarity and compute the error rate. The dataset consists of up to 1,000 English-aligned sentence pairs covering 122 languages. Following XTREME Hu et al. [2020], we directly evaluate on Tatoeba without fine-tuning and retrieve sentences with the highest cosine similarity.

**XCOPA**  In the Cross-lingual Choice of Plausible Alternatives dataset Ponti et al. [2020], each XCOPA instance corresponds to a premise and two alternatives. The task is formulated as a binary classification to predict the more plausible choice. The English COPA Gordon et al. [2012] training set and Social IQa Sap et al. [2019] training data are used for fine-tuning, while the validation and test sets of English COPA are translated and re-annotated into 11 languages for evaluation.

**MultiEURLEX**  The MultiEURLEX dataset Chalkidis et al. [2021] is a legal topic classification task that comprises 65k European Union (EU) laws in 23 official EU languages. The dataset provides multi-granular labels per document. The dataset is split into training, development, and test subsets chronologically, resulting in 55k training documents for 7 languages, and 5k each for development and test subsets in all 23 languages.

**MultiARC**  The Multilingual Amazon Reviews Corpus Keung et al. [2020] is a large-scale collection of Amazon reviews for multilingual text classification in 6 languages. Different languages are directly gathered from the marketplaces in different countries. The goal is to predict the reviewer's rating on the 5-star scale using the test of the review as input. The data is clearly split into training (200,000 reviews), development (5,000 reviews), and test sets (5,000 reviews) for each language.

**PAWS-X**  The Cross-lingual Paraphrase Adversaries from Word Scrambling Yang et al. [2019b] dataset requires identifying whether two sentences are paraphrases. A subset of the evaluation pairs in English PAWS Zhang et al. [2019] are human-translated into 6 typologically distinct languages for evaluation, while the English PAWS training set is used for training.

| Model | en | ar | bg | de | el | es | fr | hi | ru | sw | th | tr | ur | vi | zh | Avg. |
|---|---|---|---|---|---|---|---|---|---|---|---|---|---|---|---|---|
| XLM-R | 88.6 | 84.5 | 86.7 | 84.6 | 85.2 | 84.7 | 82.0 | 82.5 | 82.6 | 82.4 | 80.6 | 83.1 | 80.3 | 77.3 | 77.2 | **82.8** |
| INFOXLM | 90.4 | 83.9 | 85.8 | 86.0 | 85.6 | 87.8 | 86.9 | 83.9 | 83.5 | 83.3 | 81.2 | 84.6 | 82.7 | 81.6 | 75.7 | **84.2** |
| HICTL | 90.6 | 86.8 | 88.2 | 87.4 | 87.0 | 87.4 | 85.0 | 83.9 | 84.8 | 84.8 | 83.1 | 85.7 | 82.8 | 79.7 | 80.9 | **85.1** |
| ChatGPT | 70.4 | 61.0 | 64.5 | 64.8 | 62.8 | 65.7 | 66.3 | 51.5 | 63.4 | 55.7 | 53.0 | 61.6 | 47.9 | 61.6 | 62.6 | **60.9** |
| **EMMA-X** | **91.9** | **89.2** | **90.1** | **89.6** | **89.5** | **90.3** | **88.7** | **86.7** | **85.4** | **88.5** | **86.7** | **89.6** | **87.7** | **83.6** | **83.9** | **88.1** |

Table 10: XNLI results (accuracy) for each language.

| Model | aym | bzd | cni | gn | hch | nah | oto | quy | shp | tar | Avg. |
|---|---|---|---|---|---|---|---|---|---|---|---|
| XLM-R | 49.01 | 50.61 | 41.72 | 58.34 | 42.46 | 54.63 | 35.57 | 59.29 | 51.62 | 41.54 | **48.48** |
| INFOXLM | 49.87 | 51.29 | 42.41 | 58.83 | 43.07 | 55.25 | 36.14 | 59.87 | 52.20 | 42.12 | **49.10** |
| HICTL | 49.65 | 51.22 | 42.36 | 58.82 | 43.09 | 55.13 | 36.04 | 59.61 | 52.17 | 42.08 | **49.02** |
| ChatGPT | 42.0 | 43.6 | 40.8 | 40.4 | 40.0 | 43.8 | 41.1 | 43.1 | 42.0 | 40.0 | **41.7** |
| **EMMA-X** | **51.19** | **52.50** | **43.62** | **59.88** | **44.31** | **55.44** | **39.16** | **60.14** | **52.84** | **43.10** | **50.21** |

Table 11: AmericasNLI (ANLI) results (top-1 accuracy) across different input languages.

| Model | en-ar | en-de | en-tr | en-es | en-fr | en-it | en-nl | ar-ar | en-en | es-es | Avg. |
|---|---|---|---|---|---|---|---|---|---|---|---|
| XLM-R | 50.2 | 63.7 | 45.8 | 59.6 | 68.0 | 63.4 | 69.6 | 87.7 | 82.5 | 68.5 | **65.9** |
| INFOXLM | 81.7 | 80.3 | 79.9 | 79.1 | 80.6 | 83.4 | 81.2 | 86.7 | 87.2 | 81.7 | **82.2** |
| HICTL | 80.4 | 81.8 | 78.3 | 80.6 | 81.2 | 80.9 | 79.3 | 88.4 | 86.1 | 79.6 | **81.6** |
| **EMMA-X** | **86.6** | **85.0** | **87.1** | **84.4** | **85.2** | **89.4** | **88.3** | **90.9** | **92.0** | **84.5** | **87.3** |

Table 12: MultiSTS results (Spearman) across different input languages.

# D    Baseline Methods

To fairly evaluate the performance of EMMA-X, we choose XLM-R Conneau and Lample [2019] and its several derivatives as our baselines, which contain: (1) XLM-R, which applies multilingual MLM tasks as pre-training objectives on CCNet-100 corpus; (2) HICTL Wei et al. [2021], which continues training on XLM-R using hierarchical contrastive learning; and (3) INFOXLM, which is initialized with XLM-R and trains with cross-lingual contrast, multilingual MLM and TLM. Also, we compare EMMA-X to strong sentence models: (1) S-BERT [Reimers and Gurevych, 2020], which adopts multilingual knowledge distillation to extend monolingual sentence representations to multilingual. We use the strongest baseline, **XLM-R ← SBERT-paraphrase**, proposed in the original paper as a baseline. (2) LaBSE [Feng et al., 2022], which systematically combines several best methods, including masked language modeling, translation language modeling [Conneau and Lample, 2019], dual encoder translation ranking [Guo et al., 2018], and additive margin softmax [Yang et al., 2019a], to learn cross-lingual sentence representations. It filters 17B monolingual sentences and 6B translation pairs for sentence representation learning. We take the best model, LaBSE with Customized Vocab as our baseline. We further report the zero-shot results on Large Language Model (LLM), ChatGPT, which is trained on a wide variety of multilingual sentences and instruction tuning based on Reinforcement Learning with Human Feedback [Christiano et al., 2017, Ouyang et al., 2022].

# E    Prompts for ChatGPT

In this section, we show the input prompts of ChatGPT on each task in Table 13.

# F    Results of each Language

We show the details for tasks and all languages in Tables 10 (XNLI), 11 (AmericasNLI), 12 (Multi-STS), 14 (QE), 15 (LAReQA), 16 (Mewsli-X), 17 (XCOPA), 18 (BUCC) and 19 (PAWS-X).

## G Equations and Theoretical Analysis

### G.1 Details of Equations

**Details of Gaussian Form $\mathcal{N}_r$** In EMMA-X, GMM classifier is introduced to determine the semantic rank of sentence pairs. The posterior probability $P_{\mathcal{G}}(\cdot)$ of GMM classifier is already discussed in Eq. 5. We show the explicit calculation of Gaussian form $\mathcal{N}_r(\gamma^{(\mathbf{x}_i)}, \gamma^{(\mathbf{y}_k)})$ as:

$$\mathcal{N}_r(\gamma^{(\mathbf{x}_i)} - \gamma^{(\mathbf{y}_k)}|\mu_r, \sigma_r) = \frac{\pi_r}{(2\pi)^{(d/2)}|\text{diag}(\sigma_r)|} \cdot e^{\left(-\frac{1}{2}\left[(\gamma^{(\mathbf{x}_i)} - \gamma^{(\mathbf{y}_k)}) - \mu_r\right]^T \text{diag}(\sigma_r^{-2})\left[(\gamma^{(\mathbf{x}_i)} - \gamma^{(\mathbf{y}_k)}) - \mu_r\right]\right)}, \tag{19}$$

where $d$ is the dimension of hidden states of $\gamma^{(\mathbf{x}_i)}$ and $\gamma^{(\mathbf{y}_k)}$.

**Details of contrastive learning** The training objective of cross-lingual encoder in EMMA-X is the ranking InfoNCE loss. We show the explicit expansion of this loss (Eq. 7) as:

$$\mathcal{L}_{\mathbf{CTL}}(\mathcal{X}, \mathcal{Y}; \Theta_{\mathcal{M}}) = -\mathbb{E}_{\mathbf{x}_i \sim \mathcal{X}}\Bigg[$$

$$\log \underbrace{\frac{\sum_{\mathbf{y}_k \sim \mathcal{Y}_{c_{\mathcal{G}}^* = 1}} e^{\frac{s[\gamma^{(\mathbf{x}_i)}, \gamma^{(\mathbf{y}_k)}]}{\tau_1}}}{\sum_{\mathbf{y}_t \sim \mathcal{Y}_{c_{\mathcal{G}}^* = 1}} e^{\frac{s[\gamma^{(\mathbf{x}_i)}, \gamma^{(\mathbf{y}_t)}]}{\tau_1}} + \sum_{\mathbf{y}_t \sim \mathcal{Y}_{c_{\mathcal{G}}^* = 2}} e^{\frac{s[\gamma^{(\mathbf{x}_i)}, \gamma^{(\mathbf{y}_t)}]}{\tau_1}} + ... + \sum_{\mathbf{y}_t \sim \mathcal{Y}, c_{\mathcal{G}}^* = 4} e^{\frac{s[\gamma^{(\mathbf{x}_i)}, \gamma^{(\mathbf{y}_t)}]}{\tau_1}}}}_{\ell_1}$$

$$+ \log \underbrace{\frac{\sum_{\mathbf{y}_k \sim \mathcal{Y}_{c_{\mathcal{G}}^* = 2}} e^{\frac{s[\gamma^{(\mathbf{x}_i)}, \gamma^{(\mathbf{y}_k)}]}{\tau_2}}}{\sum_{\mathbf{y}_t \sim \mathcal{Y}_{c_{\mathcal{G}}^* = 2}} e^{\frac{s[\gamma^{(\mathbf{x}_i)}, \gamma^{(\mathbf{y}_t)}]}{\tau_2}} + \sum_{\mathbf{y}_t \sim \mathcal{Y}_{c_{\mathcal{G}}^* = 3}} e^{\frac{s[\gamma^{(\mathbf{x}_i)}, \gamma^{(\mathbf{y}_t)}]}{\tau_2}} + \sum_{\mathbf{y}_t \sim \mathcal{Y}_{c_{\mathcal{G}}^* = 4}} e^{\frac{s[\gamma^{(\mathbf{x}_i)}, \gamma^{(\mathbf{y}_t)}]}{\tau_2}}}}_{\ell_2} \tag{20}$$

$$+ \log \underbrace{\frac{\sum_{\mathbf{y}_k \sim \mathcal{Y}_{c_{\mathcal{G}}^* = 3}} e^{\frac{s[\gamma^{(\mathbf{x}_i)}, \gamma^{(\mathbf{y}_k)}]}{\tau_3}}}{\sum_{\mathbf{y}_t \sim \mathcal{Y}_{c_{\mathcal{G}}^* = 3}} e^{\frac{s[\gamma^{(\mathbf{x}_i)}, \gamma^{(\mathbf{y}_t)}]}{\tau_3}} + \sum_{\mathbf{y}_t \sim \mathcal{Y}_{c_{\mathcal{G}}^* = 4}} e^{\frac{s[\gamma^{(\mathbf{x}_i)}, \gamma^{(\mathbf{y}_t)}]}{\tau_3}}}}_{\ell_3}\Bigg],$$

where $\tau_r$ represents the temperature term. As small temperature $\tau$ tends to be less tolerant to similar samples, and large $\tau$ tends to cluster similar samples together [Wang and Liu, 2021], we empirically set $\tau_1 < \tau_2 < \tau_3 < \tau_4$, which remains the same as Hoffmann et al. [2022].

### G.2 Theoretical Analysis

In this section, we provide detailed proof for Eq. 14 and Eq. 15. Next, we prove the feasibility of our dual supervision. GMM classifier clusters sentence pairs in terms of Euclidean distance, while cross-lingual encoder minimizes the covariance of each semantic relation rank via cosine distance. Finally, we prove that these two metrics are actually equivalent to each other in the unit hypersphere of the embedding space.

**Proof of Eq. 14.** We provide the derivation of Eq. 14. With the assumption that $P(\mathbf{x}_i, \mathbf{y}_k | c_{\mathcal{G}}^* = r, \Theta) \sim \mathcal{N}_r(\mathbf{x}_i - \mathbf{y}_k | \tilde{\mu}_r, \tilde{\sigma}_r)$, we have,

$$
\sum_{\mathbf{x}_i \in \mathcal{X}} \sum_{\mathbf{y}_k \in \mathcal{Y}} \sum_{r=1}^{N} Q(r) \log \frac{P(\mathbf{x}_i, \mathbf{y}_k, r | \Theta)}{Q(r)} \approx \sum_{\mathbf{x}_i \in \mathcal{X}} \sum_{\mathbf{y}_k \in \mathcal{Y}} \sum_{r=1}^{N} \log P(\mathbf{x}_i, \mathbf{y}_k | c_{\mathcal{G}}^* = r, \Theta)
$$

$$
= \sum_{\mathbf{x}_i \in \mathcal{X}} \sum_{\mathbf{y}_k \in \mathcal{Y}} \sum_{r=1}^{N} \Big( \log \big( \frac{1}{(2\pi)^{(d/2)} |\tilde{\sigma}_r|^{1/2}} \big)
$$

$$
+ \frac{1}{2} \big[ (\mathbf{x}_i - \mathbf{y}_k) - \tilde{\mu}_r \big]^T \tilde{\sigma}_r^{-1} \big[ (\mathbf{x}_i - \mathbf{y}_k) - \tilde{\mu}_r \big] \Big)
$$

$$
\geq \sum_{r=1}^{N} \big[ \sum_{\mathbf{x}_i \in \mathcal{X}} \sum_{\mathbf{y}_k \in \mathcal{Y}} (\mathbf{x}_i - \mathbf{y}_k) \big]^2 - 2\tilde{\mu}_r \sum_{\mathbf{x}_i \in \mathcal{X}} \sum_{\mathbf{y}_k \in \mathcal{Y}} (\mathbf{x}_i - \mathbf{y}_k) + n\tilde{\mu}_r^2
$$

$$
= \sum_{r=1}^{N} n^2 \tilde{\mu}_r^2 - n\tilde{\mu}_r^2
$$

$$
= n(n-1) \sum_{r=1}^{N} \tilde{\mu}_r^2,
$$

$$(21)$$

with $n$ denoting the number of sentence pairs in semantic rank $r$. Here, we ignore the impact of $\tilde{\sigma}_r$.

**Proof of Eq. 15.** As we apply dual supervision, data in the contrastive label space also follows the distribution $\mathcal{N}_r(\mathbf{x}_i - \mathbf{y}_k | \tilde{\mu}_r, \tilde{\sigma}_r)$. Hence, under mild assumptions, we can get:

$$
\mathcal{L}_{\mathbf{CTL}}^+(\mathcal{X}, \mathcal{Y}; \Theta_{\mathcal{M}}) = \mathbb{E}_{\mathbf{x}_i \sim \mathcal{X}} \sum_{r=1}^{N-1} \log \sum_{\mathbf{y}_k \sim \mathcal{Y}_{c_{\mathcal{G}}^* = r}} e^{s[\gamma^{(\mathbf{x}_i)}, \gamma^{(\mathbf{y}_k)}]}
$$

$$
= \sum_{\mathbf{x}_i \in \mathcal{X}} \sum_{\mathbf{y}_k \in \mathcal{Y}} \sum_{r=1}^{N-1} s(\mathbf{x}_i, \mathbf{y}_k)
$$

$$
= \sum_{\mathbf{x}_i \in \mathcal{X}} \sum_{\mathbf{y}_k \in \mathcal{Y}} \sum_{r=1}^{N-1} \frac{(\mathbf{x}_i - \mathbf{y}_k)^2 - 2}{2}
$$

$$
= n^2 \sum_{r=1}^{N-1} \tilde{\mu}_r^2.
$$

$$(22)$$

Based on the definition of semantic ranks, we have $\tilde{\mu}_1 < \tilde{\mu}_2 < ... < \tilde{\mu}_N$. Empirically, the number of sentence pairs in each rank $n$ is larger than the number of semantic ranks $N$. Hence, it can be derived that:

$$
\mathcal{L}_{\mathbf{CTL}}^+(\mathcal{X}, \mathcal{Y}; \Theta_{\mathcal{M}}) = n^2 \sum_{r=1}^{N-1} \tilde{\mu}_r^2
$$

$$
< n^2 \sum_{r=1}^{N-1} \tilde{\mu}_r^2 + n^2 \tilde{\mu}_N^2 - n \sum_{r=1}^{N} \tilde{\mu}_r^2
$$

$$
= n(n-1) \sum_{r=1}^{N} \tilde{\mu}_r^2
$$

$$
\leq \sum_{\mathbf{x}_i \in \mathcal{X}} \sum_{\mathbf{y}_k \in \mathcal{Y}} \sum_{r=1}^{N} Q(r) \log \frac{P(\mathbf{x}_i, \mathbf{y}_k, r | \Theta)}{Q(r)}.
$$

$$(23)$$

Therefore, we prove that minimizing the positive terms $\mathcal{L}_{\mathbf{CTL}}^+(\mathcal{X}, \mathcal{Y}; \Theta_{\mathcal{M}})$ in contrastive learning is equivalent to maximizing a lower bound of the likelihood in Eq. 12.

According to the definition of semantic ranks, the approximated semantic rank $c_{\mathcal{G}}^*$ from GMM classifier should satisfy the following restriction,

$$
\mathbb{E}_{\mathbf{y}_k \sim \mathcal{Y}_{c_{\mathcal{G}}^* = 1}} ||\gamma^{(\mathbf{x}_i)} - \gamma^{(\mathbf{y}_k)}|| < \mathbb{E}_{\mathbf{y}_k \sim \mathcal{Y}_{c_{\mathcal{G}}^* = 2}} ||\gamma^{(\mathbf{x}_i)} - \gamma^{(\mathbf{y}_k)}|| < ... < \mathbb{E}_{\mathbf{y}_k \sim \mathcal{Y}_{c_{\mathcal{G}}^* = N}} ||\gamma^{(\mathbf{x}_i)} - \gamma^{(\mathbf{y}_k)}||. \quad (24)
$$

Similarly, the approximated semantic rank $c_{\mathcal{M}}^*$ from cross-lingual encoder should satisfy the following restriction,

$$\mathbb{E}_{\mathbf{y}_k \sim \mathcal{Y}_{c_{\mathcal{M}}^*=1}} s[\gamma^{(\mathbf{x}_i)}, \gamma^{(\mathbf{y}_k)}] > \mathbb{E}_{\mathbf{y}_k \sim \mathcal{Y}_{c_{\mathcal{M}}^*=2}} s[\gamma^{(\mathbf{x}_i)}, \gamma^{(\mathbf{y}_k)}] > ... > \mathbb{E}_{\mathbf{y}_k \sim \mathcal{Y}_{c_{\mathcal{M}}^*=N}} s[\gamma^{(\mathbf{x}_i)}, \gamma^{(\mathbf{y}_k)}]. \quad (25)$$

Next, we prove that these two restrictions are interchangeable with each other in a unit hypersphere. For simplicity, we consider only two ranks, but extending the explanation to more ranks is trivial. As the Euclidean distance is always larger than 0, we have:

$$\mathbb{E}_{\mathbf{y}_k \sim \mathcal{Y}_{c_{\mathcal{G}}^*=1}} ||\gamma^{(\mathbf{x}_i)} - \gamma^{(\mathbf{y}_k)}|| < \mathbb{E}_{\mathbf{y} \sim \mathcal{Y}_{c_{\mathcal{G}}^*=2}} ||\gamma^{(\mathbf{x}_i)} - \gamma^{(\mathbf{y}_k)}||$$

$$\Leftrightarrow \mathbb{E}_{\mathbf{y}_k \sim \mathcal{Y}_{c_{\mathcal{G}}^*=1}} (\gamma^{(\mathbf{x}_i)} - \gamma^{(\mathbf{y}_k)})^2 < \mathbb{E}_{\mathbf{y}_k \sim \mathcal{Y}_{c_{\mathcal{G}}^*=2}} (\gamma^{(\mathbf{x}_i)} - \gamma^{(\mathbf{y}_k)})^2$$

$$\Leftrightarrow \mathbb{E}_{\mathbf{y}_k \sim \mathcal{Y}_{c_{\mathcal{G}}^*=1}} (2 - 2\gamma^{(\mathbf{x}_i)}\gamma^{(\mathbf{y}_k)}) < \mathbb{E}_{\mathbf{y}_k \sim \mathcal{Y}_{c_{\mathcal{G}}^*=2}} (2 - 2\gamma^{(\mathbf{x}_i)}\gamma^{(\mathbf{y}_k)}) \quad (26)$$

$$\Leftrightarrow \mathbb{E}_{\mathbf{y}_k \sim \mathcal{Y}_{c_{\mathcal{G}}^*=1}} s[\gamma^{(\mathbf{x}_i)}, \gamma^{(\mathbf{y}_k)}] > \mathbb{E}_{\mathbf{y}_k \sim \mathcal{Y}_{c_{\mathcal{G}}^*=2}} s[\gamma^{(\mathbf{x}_i)}, \gamma^{(\mathbf{y}_k)}]$$

$$\Leftrightarrow \mathbb{E}_{\mathbf{y}_k \sim \mathcal{Y}_{c_{\mathcal{M}}^*=1}} s[\gamma^{(\mathbf{x}_i)}, \gamma^{(\mathbf{y}_k)}] > \mathbb{E}_{\mathbf{y}_k \sim \mathcal{Y}_{c_{\mathcal{M}}^*=2}} s[\gamma^{(\mathbf{x}_i)}, \gamma^{(\mathbf{y}_k)}].$$

From the above analyses, we can tell that the approximated semantic rank from one module can provide a reasonable supervision signal to guide the training of the other module. Hence, all sentence pairs will be uniformly distributed according to a unified ranking semantic similarity in the embedding space.

**Basic Prompt for XNLI/ANLI**

**Task Description:** Read the following and determine the relationship between Hypothesis and Premise. Choose relation from "contradiction", "neutral", or "entailment".

**Hypothesis:** Yo... no puedo pensar por qué deberías hablarme así, dijo ella, con menos de lo que le había asegurado antes.

**Premise:** Ella era una buena amiga de él, por esto le dolía que le hablara así.

**Basic Prompt for MultiSTS**

**Task Description:** Read the following sentences and measure the real-valued meaning similarity between these two sentences. You can choose the meaning similarity score, ranging from 0 for no meaning overlap to 5 for meaning equivalence.

**Sentence1:** A person is on a baseball team.

**Sentence2:** Eine Person spielt in einem Team Basketball.

**Basic Prompt for QE**

**Task Description:** Read the Source sentence and its Translation, and estimate the quality of the Translation. You can rate the translation from 0-1 according to the perceived translation quality.

**Source:** În Franța a început stagnarea demografică de lungă durată, refacerea durând o generație.

**Translation:** In France, long-term demographic stagnation has started, restoring a generation.

**Basic Prompt for XCOPA**

**Task Description:** Read the Premise and determine which choice is the effect(or cause) of the Premise . Choose from "Choice1" or "Choice2".

**Premise:** Kuki kurukuna wasiman haykurqanku.

**Choice1:** Kuki kurukunaqa wasimanta chinkarqanku.

**Choice2:** Kuki kuruqa wasip kurkunta mikhurqanku.

**Basic Prompt for MultiEURLEX**

**Task Description:** Read the following sentences and determine the legal topic of the given sentence. The legal topic should choose from 'international organisations', 'social questions', 'production', 'technology and research', 'environment', 'energy', 'transport', 'law', 'finance', 'education and communications', 'trade', 'agriculture', 'forestry and fisheries', 'economics', 'agri-foodstuffs', 'EUROPEAN UNION', 'science', 'politics', 'international relations', 'industry', 'geography', 'business and competition', 'employment and working conditions'.

**Sentence:** NEUVOSTON ASETUS (EU) N:o 1390/2013, annettu 16 päivänä joulukuuta 2013, Euroopan unionin ja Komorien liiton kesken näiden välisessä kalastuskumppanuussopimuksessa määrätyjen kalastusmahdollisuuksien ja taloudellisen korvauksen vahvistamisesta hyväksytyn pöytäkirjan mukaisten kalastusmahdollisuuksien jakamisesta ...

**Basic Prompt for MultiARC**

**Task Description:** Read the following review and predict a 5-star scale rating (1 means the poorest experience and 5 represents excellent or outstanding performance) that can best match the review.

**Review:** no me llego el articulo me lo mando por correos normal sin seguimiento y nunca me llego tota un desastre

**Basic Prompt for PAWS-X**

**Task Description:** Read the following sentences and determine whether two sentences are paraphrases. Return yes or no.

**Sentence1:** La excepción fue entre fines de 2005 y 2009 cuando jugó en Suecia con Carlstad United BK, Serbia con FK Borac Čačak y el FC Terek Grozny de Rusia.

**Sentence2:** La excepción se dio entre fines del 2005 y 2009, cuando jugó con Suecia en el Carlstad United BK, Serbia con el FK Borac Čačak y el FC Terek Grozny de Rusia.

Table 13: Prompts of ChatGPT on each task.

| Model | en-de | en-zh | et-en | ne-en | ro-en | ru-en | si-en | en-cs | en-ja | km-en | ps-en | Avg. |
|---|---|---|---|---|---|---|---|---|---|---|---|---|
| XLM-R | 0.412 | 0.566 | 0.797 | 0.812 | 0.891 | 0.774 | 0.578 | 0.547 | 0.335 | 0.612 | 0.635 | **0.632** |
| INFOXLM | 0.517 | 0.534 | 0.775 | 0.834 | 0.890 | 0.788 | 0.581 | 0.564 | 0.325 | 0.635 | 0.616 | **0.641** |
| HICTL | 0.495 | 0.579 | 0.792 | 0.835 | **0.904** | 0.787 | 0.575 | 0.556 | 0.342 | 0.625 | 0.648 | **0.649** |
| **EMMA-X** | **0.580** | **0.589** | **0.809** | **0.854** | 0.897 | **0.829** | **0.593** | **0.577** | **0.370** | **0.641** | **0.651** | **0.672** |

Table 14: WMT21-QE-Task1 results (Pearson) across different input languages.

| Model | ar | de | el | en | es | hi | ru | th | tr | vi | zh | Avg. |
|---|---|---|---|---|---|---|---|---|---|---|---|---|
| XLM-R | 34.1 | 42.4 | 39.3 | 44.8 | 44.0 | 37.3 | 41.7 | 38.6 | 40.9 | 40.4 | 39.5 | **40.3** |
| INFOXLM | 39.7 | 52.6 | 39.2 | 55.1 | 53.4 | 36.8 | 51.0 | 28.5 | 41.1 | 48.9 | 47.3 | **44.9** |
| HICTL | 40.3 | 53.2 | 41.7 | 56.3 | 54.3 | 39.6 | 51.7 | 30.1 | 42.8 | 48.9 | 48.5 | **46.1** |
| **EMMA-X** | **45.1** | **58.4** | **45.4** | **60.6** | **59.8** | **41.4** | **56.3** | **34.7** | **47.1** | **54.6** | **53.4** | **50.6** |

Table 15: LAReQA results (mean average precision@20, mAP@20) across different input languages.

| Model | ar | de | en | es | fa | ja | pl | ro | ta | tr | uk | Avg. |
|---|---|---|---|---|---|---|---|---|---|---|---|---|
| XLM-R | 34.6 | 66.0 | 62.6 | 64.8 | 27.1 | 47.8 | 64.8 | 33.7 | 17.8 | 62.3 | 53.2 | **48.6** |
| INFOXLM | 40.8 | 71.6 | 66.3 | 68.7 | 48.7 | 61.0 | 66.7 | 39.2 | 42.0 | 64.6 | 58.1 | **57.1** |
| HICTL | 41.7 | 68.5 | 64.2 | 65.6 | 45.6 | 51.9 | 67.6 | 40.4 | 32.8 | 65.5 | 58.9 | **54.8** |
| **EMMA-X** | **50.2** | **78.7** | **69.1** | 63.7 | 47.9 | **59.6** | **70.0** | **50.2** | **43.5** | **68.0** | **60.9** | **59.6** |

Table 16: Mewsli-X results (mean average precision@20, mAP@20) across different input languages.

| Model | et | ht | id | it | qu | sw | ta | th | tr | vi | zh | Avg. |
|---|---|---|---|---|---|---|---|---|---|---|---|---|
| XLM-R | 73.8 | 67.4 | 77.8 | 72.2 | 52.3 | 70.9 | 72.1 | **74.6** | 73.4 | 73.2 | 75.7 | **71.2** |
| INFOXLM | 75.1 | 73.4 | **78.3** | 80.7 | 65.6 | 69.1 | 72.7 | 73.9 | 76.9 | 77.8 | 77.5 | **74.6** |
| HICTL | 75.9 | 73.1 | 77.8 | **81.2** | 65.5 | 73.8 | 72.6 | 73.2 | 76.1 | 75.4 | 78.0 | **74.8** |
| ChatGPT | 80.6 | 64.1 | 85.6 | 89.2 | 47.4 | 75.9 | 56.4 | 67.3 | 82.2 | 81.5 | 85.8 | **74.2** |
| **EMMA-X** | **76.8** | **74.0** | 77.6 | 79.8 | **76.2** | **74.4** | **74.4** | 74.2 | **77.6** | **82.6** | **89.6** | **78.2** |

Table 17: XCOPA results (accuracy) across different input languages.

| Model | de | fr | ru | zh | Avg. |
|---|---|---|---|---|---|
| XLM-R | 76.1 | 72.3 | 62.3 | 60.8 | **67.9** |
| INFOXLM | 81.3 | 78.2 | 76.0 | 74.2 | **77.4** |
| HICTL | 80.5 | 79.2 | 76.0 | 74.8 | **77.6** |
| **EMMA-X** | **85.1** | **82.8** | **81.3** | **78.3** | **81.9** |

Table 18: BUCC results (F1) across different languages.

| Model | en | de | es | fr | ja | ko | zh | Avg. |
|---|---|---|---|---|---|---|---|---|
| XLM-R | 95.7 | 92.2 | 92.7 | 92.5 | 84.7 | 85.9 | 87.1 | **90.1** |
| INFOXLM | **97.7** | 94.6 | **95.2** | 95.1 | 88.9 | 89.0 | 90.2 | **93.0** |
| HICTL | 97.4 | 94.2 | 95.0 | 94.2 | 89.1 | 89.5 | 90.2 | **92.8** |
| ChatGPT | 71.9 | 67.8 | 67.9 | 67.0 | 58.3 | 54.7 | 61.4 | **64.2** |
| **EMMA-X** | 97.3 | **95.6** | 94.7 | **96.0** | **92.9** | **89.8** | **93.0** | **94.2** |

Table 19: PAWS-X results (accuracy) for each language.

