# OpenReview forum: "EMMA-X: An EM-like Multilingual Pre-training Algorithm for Cross-lingual Representation Learning"
_NeurIPS.cc/2023/Conference — NeurIPS 2023 poster_

### Official Review · Reviewer_tN44 · 2023-06-27

**Soundness:** 2 fair
**Presentation:** 3 good
**Contribution:** 3 good
**Rating:** 5
**Confidence:** 5

**Summary:**

This paper proposed EM-like multilingual pre-training algorithm using both supervised parallel data and unsupervised monolingual data to train an encoder model such that it can produce language agnostic representations for multiple languages. That is, embedding space of sentences of similar meanings will be the same across languages. The method uses an EM procedures that employs an GMM classifier, which is pre-trained with parallel data to produce a similarity rank between 2 input sentence, and then train the encoder model both with parallel data and ranks produced by GMM with unsupervised data.
The method achieves higher scores across many inference, similarity and retrieval tasks compared to unsupervised XLM-R.

**Strengths:**

* The methodology seems original and new, especially for the new rank-based classifier.
* The paper presentation and method description is clear and understandable.

**Weaknesses:**

There are multiple concerns with the evaluation and comparison with the method.

* Despite utilize large-scale parallel data from multiple languages heavily in the pretraining stage of encoder and GMM classififer, the paper compares its method with unsupervised models like XLM-R. This make the comparison in main table 1 invalid and incomparable. This method is more semi-supervised with supervised pretraining and unsupervised finetuning stages.
* There are some comparison with supervised models like LaBSE, but apparently the proposed method do not score well with supervised models. Moreover, the writting seems to suggest the paper is comparing unsupervised vs supervised but in fact this is supervised vs supervised.
* XLM-R is not designed for language agnostic representation, but there are notable baselines that achieve the same language-agnostic object with simpler methodology: [LASER](https://github.com/facebookresearch/LASER) which use parallel data and [CRISS](https://arxiv.org/abs/2006.09526) which does not use parallel data. Not only they are not compared, they are also not mentioned in the paper.

**Questions:**

NA

**Limitations:**

No limitation mentioned.

---

> ### Author Rebuttal · Authors · 2023-08-08
>
> Thanks for your comments.
>
> **Q1: Unfair comparison between EMMA-X and unsupervised method (XLM-R).**
>
> As a semi-supervised method, we have undertaken a comprehensive evaluation by comparing our results with four supervised methods (InfoXLM [1], HiCTL [2], LaBSE [3], and S-BERT [4]) in Tables 1, 2, and 3, leading to substantial improvements. The reason for including XLM-R [5] in our comparisons is that EMMA-X is initialized from XLM-R. **To ensure a fair assessment, we additionally retrained XLM-R, InfoXLM, HiCTL using the same parallel data as EMMA-X, and EMMA-X can outperform retrained baselines by 7.97% on average as presented in Table 1 with the symbol \ddag.**
>
> **Q2: Poor performance on EMMA-X when compared with supervised models (LaBSE).**
>
> We have not claimed that EMMA-X is an unsupervised method and explicitly emphasize the usage of supervised data to initialize the model in both Section 3.2 and Algorithm 1. As you said, EMMA-X is a semi-supervised method. EMMA-X can outperform two supervised baselines on all tasks and even outperform the state-of-the-art supervised baseline (LaBSE) on two tasks and all long-tail languags. It is hard to directly compare EMMA-X with LaBSE, as LaBSE uses a large pre-training dataset containing CC-100, Wikipedia, and a fine-selected 6B translation corpus that are much better than EMMA-X in both quantity and quality. However, the notable improvements exhibited by EMMA-X on WMT21QE [6], Mewsli-X [7], and LaREQA [8] in Table 2 and on low-resource languages of Tatoeba [9] and FLORES-200 [10] in Table 3 can further demonstrate the effectiveness of EMMA-X.
>
> **Q3: Lacking baselines and citations.**
>
> While LASER [11] is a notable baseline, we do not compare with it because we have chosen a stronger and more recent baseline, LaBSE. We have cited LASER in our paper from the perspective of its data **in line 235**, and we will add a more comprehensive discussion about LASER, LaBSE, S-BERT, and EMMA-X in the next version of our paper.
> EMMA-X and its baselines are encoder-only model designed for learning representations, while CRISS [12] is an unsupervised seq2seq model that mainly performs on machine translation. We will mention CRISS in the next version.
>
> **References:**
>
> [1] Chi Z, Dong L, Wei F, et al. InfoXLM: An information-theoretic framework for cross-lingual language model pre-training.
>
> [2] Wei X, Weng R, Hu Y, et al. On learning universal representations across languages.
>
> [3] Feng F, Yang Y, Cer D, et al. Language-agnostic BERT sentence embedding.
>
> [4] Reimers N, Gurevych I. Sentence-bert: Sentence embeddings using siamese bert-networks.
>
> [5] Conneau A, Khandelwal K, Goyal N, et al. Unsupervised cross-lingual representation learning at scale.
>
> [6] Lucia Specia, Frédéric Blain, Marina Fomicheva, Chrysoula Zerva, Zhenhao Li, Vishrav Chaudhary, and André F. T. Martins. Findings of the WMT 2021 shared task on quality estimation.
>
> [7] Sebastian Ruder, Noah Constant, Jan Botha, Aditya Siddhant, Orhan Firat, Jinlan Fu, Pengfei Liu, Junjie Hu, Dan Garrette, Graham Neubig, and Melvin Johnson. XTREME-R: Towards more challenging and nuanced multilingual evaluation.
>
> [8] Uma Roy, Noah Constant, Rami Al-Rfou, Aditya Barua, Aaron Phillips, and Yinfei Yang. LAReQA: Language-agnostic answer retrieval from a multilingual pool.
>
> [9] Mikel Artetxe and Holger Schwenk. Massively multilingual sentence embeddings for zero-shot cross-lingual transfer and beyond.
>
> [10] Marta R Costa-jussà, James Cross, Onur Çelebi, Maha Elbayad, Kenneth Heafield, Kevin Heffernan, Elahe Kalbassi, Janice Lam, Daniel Licht, Jean Maillard, et al. No language left behind: Scaling human-centered machine translation.
>
> [11] Mikel Artetxe and Holger Schwenk. Massively multilingual sentence embeddings for zero-shot cross-lingual transfer and beyond.
>
> [12] Tran C, Tang Y, Li X, et al. Cross-lingual retrieval for iterative self-supervised training.

---

> > ### Comment · Reviewer_tN44 · 2023-08-14
> > **Response**
> >
> > Thanks the authors for the clarification.
> >
> > Given it is a fair comparison, the method is still not strong and outperforms consistently across benchmarks. Nonetheless I changed the scores to award the novelty of the work.

---

> > > ### Author Response · Authors · 2023-08-14
> > >
> > > We sincerely appreciate your valuable feedback on our paper. Your suggestions have been helpful in guiding our revisions. We will take your advice into consideration and include more thorough comparisons between EMMA-X and baselines, which will serve to validate and enhance our research findings. Your guidance and support are highly appreciated. Thank you for helping us improve our work.

---

### Official Review · Reviewer_oxNi · 2023-06-27

**Soundness:** 3 good
**Presentation:** 3 good
**Contribution:** 3 good
**Rating:** 7
**Confidence:** 4

**Summary:**

This paper proposes an EM-like pre-training algorithm called EMMA-X to learn cross-lingual representations. The authors unify semantic relation classification and universal representation learning into the framework. To this end, a GMM classifier and a cross-lingual encoder is jointly trained in the algorithm. Finally, the authors validate their method by conducting experiments on 12 cross-lingual tasks including inference, similarity, retrieval, and classification.

**Strengths:**

- The paper is well-structured and easy to follow.
- The EM-like pre-training, which includes a semantic relation rank task, is somewhat novel.
- The experiments are extensive, and the results generally indicate the proposed method is effective.
- The geometric analysis section is interesting and informative, where the authors visualize the representations from four different models.

**Weaknesses:**

- Although the authors claim that EMMA-X can learn good representations with excessive non-parallel, the encoder, as far as I can see, is continue-trained on parallel sentence pairs with InfoNCE on parallel corpora for initialization. Therefore, this initialization weakens the claim.

- The author does not verify the choice of a GMM classifier. At least, the author should give some simple intuition to the readers.

- Semantic relation task is very important in the proposed framework. The authors claim that a binary separation is not good, but do not give further justification for the choices of N=4 semantic ranks.


**Questions:**

- Consider removing the comma after each equation
- Line 69: Why do we have to use L-2 normalization? There are multiple ways to perform normalizations, L-2 is just one of them. Therefore, I would rather say $f(\cdot)$ is a normalization, e.g., L-2.
- Figure 1 in caption: "{y1, and y2, y3, ...}" -> {$y\_1, y\_2, y\_3, \cdots$}
- In Figure 1, I assume $\gamma^{(x)} - \gamma^{(y_1)}$ is missing
- The number of semantic ranks is set to 4. This sounds to me like a random selection. Did you do experiments to justify the choice?
- The paper claims that the model can learn to learn cross-lingual universals with the aid of massive multilingual non-parallel data, but the cross-lingual encoder is continued trained on infoNCE on some parallel sentence pairs. I would assume this continue-training step is even more important than the method proposed. I would suggest the author justify the initialization by simply using the model weights of XLM-R.
- Line 188-190: The number of sentence pairs is 3.2 billion, but what is the exact number of non-parallel sentences? That should also be mentioned in the main content of the paper.
- Line 192: $\mathbb{R}^1$ -> $\mathbb{R}$
- Table 1: the authors should also use special characters to denote which results (BUCC and Tatoeba?) are obtained by zero-shot for EMMA-X.
- Figure 2 and Table 4: according to the results, S-BERT performs very well (even better) than the proposed method, especially on long-tail languages. In this case, what would be the benefit of applying EMMA-X to those languages instead of simply using S-BERT?


**Limitations:**

The paper does not include a limitation or broader impact section, therefore not applicable.

---

> ### Author Rebuttal · Authors · 2023-08-08
>
> Thanks for your positive comments.
>
> **Q1: Initialization with parallel corpora weakens motivation.**
>
> The primary goal of EMMA-X is to acquire universal semantic representations for a multitude of languages. However, it is important to note that only a limited number of languages (4%) possess parallel data. Therefore, EMMA-X strives to leverage non-parallel data to extend language coverage for universal semantic representations and to enhance representation performance for those languages that have limited available resources.
> To provide additional evidence supporting our claim, we propose an ablation experiment in the General response to all authors. The results in Table 5 demonstrate that EMMA-X significantly improves the retrieval accuracy by 8.1 when compared to solely using parallel data. Moreover, on 16 long-tail languages, the retrieval accuracy further increases from 76.2 to 90.7, representing a substantial improvement of 19.0%. These findings support our claim that EMMA-X can indeed learn effective universal semantic representations from an abundance of non-parallel data.
>
> **Q2: Comparison between GMM classifier and other simple ones.**
>
> Please refer to the response to Q2 included in the “**General responses to all Reviewers**” part for details.
>
> **Q3: Missing justification for the choice of semantic similarity ranks.**
>
> Please refer to the response to Q3 included in the “**General responses to all Reviewers**” part for details.
>
> **Q4: The reason to use L2-Normalization.**
>
> We measure the semantic similarity between sentence via cosine similarity, which requires an L2-normalization to the output of embeddings of encoder. We also apply L2-Normalization following the common configuration in MoCo [1], HiCTL [2], and InfoXLM [3].
>
> **Q5: The exact number of non-parallel sentences.**
>
> As shown in Appendix A, we use 800B of non-parallel data covering 94 languages, which is about 67 times larger than the parallel data. We will add the statistics to the main content of the paper in the next version.
>
> **Q6: The benefit of using EMMA-X rather than S-BERT, based on results from Figure 2 and Table 4.**
>
> Table 4 demonstrates that EMMA-X exhibits superior performance on low-resource languages compared to S-BERT [4], surpassing it by 69.5%, 5.0%, and 29.5% in three geometric criteria, respectively. Similarly, Figure 2 provides visual evidence supporting this claim. Specifically, EMMA-X showcases smaller language-specific clusters on four low-tail languages (tk, xh, tt, and eo) when compared to S-BERT.
>
> The abnormal results observed on the high KL-divergence, as depicted in Table 4 and Figure 2(k), can be attributed to the representation space for S-BERT deteriorating into a low-dimensional manifold (low isotropy score in Table 4 and Figure 2k), and different languages are not distributed uniformly across the whole representation space, which limits the expressive ability.
>
> **Q7: Typo and expression errors.**
>
> Thanks for pointing out the errors. We will proofread the content again to fix the wrong expressions and typos.
>
> **References:**
>
> [1] He K, Fan H, Wu Y, et al. Momentum contrast for unsupervised visual representation learning.
>
> [2] Wei X, Weng R, Hu Y, et al. On learning universal representations across languages.
>
> [3] Chi Z, Dong L, Wei F, et al. InfoXLM: An information-theoretic framework for cross-lingual language model pre-training.
>
> [4] Reimers N, Gurevych I. Sentence-bert: Sentence embeddings using siamese bert-networks.

---

> > ### Comment · Reviewer_oxNi · 2023-08-18
> >
> > Thank you very much for your detailed response. All my questions have been appropriately answered now. I'll raise the recommendation score.

---

> > > ### Author Response · Authors · 2023-08-18
> > >
> > > Thank you so much for your thoughtful message. I'm delighted to know that I could assist you in providing the information you needed. Your willingness to raise the recommendation score is truly generous and means a lot to me. Thanks again for your appreciation!

---

### Official Review · Reviewer_s5pr · 2023-07-06

**Soundness:** 2 fair
**Presentation:** 3 good
**Contribution:** 3 good
**Rating:** 7
**Confidence:** 4

**Summary:**

This paper proposes a new approach to learn cross-lingual representation learning as a universal alignment solution for any two languages without parallel data. The proposed approach resembles EM-like algorithm, which consists of a classifier to quantify the semantic similarity of two non-parallel sentences, as well as a sentence encoder that aligns the embedding representation of different languages. During the training process, the two components receive supervision from each other to reinforce the capability to recognize the accurate semantic similarity, where the paper also provides the theoretical justification to demonstrate the mutual influence being positive and effective. A new benchmark is also proposed that comprises four existing cross-lingual transfer tasks; the proposed approach is compared against other recent baselines, as well as ChatGPT. The empirical results show that new state-of-the-art performance can be obtained by this new approach with good margins.

**Strengths:**

- A new approach is proposed based on a novel EM-like algorithm with mutual inference and supervision by two components. Especially, the approach proposes to rank the similarity of a sentence pair, rather than optimizing the traditional binary positive/negative contrastive objective, which is indeed interesting and effective shown by the results.

- State-of-the-art performance is achieved by this approach on the new benchmark consisting different types of existing cross-lingual transfer tasks, comparing against strong baselines such as XLM-R.

- A theoretical analysis is provided, in addition to the superior empirical results to justify the effectiveness of the proposed approach.

**Weaknesses:**

- It might be over-claiming that the proposed approach operates to align the representation of different languages without parallel data, since Algorithm 1 clearly shows that parallel corpus is needed to bootstrap both the similarity classifier and the sentence encoder, and all experiments are based on this warm up that leverages parallel corpus.

- There lacks more in-depth analysis regarding the ablation of the proposed approach. In particular:

  - There could be a simplified version of this approach without iterative optimization: we can first train a similarity classifier based on the proposed mixed-up method, and generate similarity rankings of sentence pairs on non-parallel data, which can then be used to optimize the contrastive objective directly for training the encoder. In this way, we can verify whether the improvement mostly comes from the fine-grained similarity ranking, and how much this iterative optimization mechanism contributes to the final performance.

  - There could also be another experiment that downgrades the similarity ranking to binary, as if the traditional positive/negative pair. Thus, it could show again that whether this EM-like algorithm is really necessary and how much this process could improve upon the plain contrastive objective.

**Questions:**

What is the primary reason for adopting GMM as the similarity classifier? Can we use a normal classifier (e.g. using CLS on Transformers) and simply use their softmax probability for each rank?

**Limitations:**

It would be good to discuss how much degradation could happen if there is not enough parallel corpus for the warm-up.

---

> ### Author Rebuttal · Authors · 2023-08-08
>
> Thanks for your constructive reviews.
>
> **Q1: Over-claiming EMMA-X operates without parallel data.**
>
> The primary goal of EMMA-X is to acquire universal semantic representations for a multitude of languages. However, it is important to note that only a limited number of languages (4%) possess parallel data. Therefore, EMMA-X strives to leverage non-parallel data to extend language coverage for universal semantic representations and to enhance representation performance for those languages that have limited available resources.
>
> **Q2: Adding an ablation study about the EM iterative optimization.**
>
> Thanks for your constructive advice. We apply an iterative paradigm to optimize the GMM classifier since the parallel data is not sufficient to cover so many languages to train a classifier and the semantic rank expectation from GMM classifier can be biased. The iterative optimization can help GMM classifier to learn better on long-tail languages.
> To provide additional evidence supporting our claim, we compare EMMA-X with a new baseline that applies a fixed GMM classifier to generate similarity ranks to optimize the encoder, referred to as "Phase 1 + Fixed GMM". The results are shown in the ablation experiments (Table 5) in "General Response to All Reviewers." From Table 5, we can see that EMMA-X improves retrieval accuracy by 6.6 compared with "Phase 1 + Fixed GMM". On 16 long-tail languages, the retrieval accuracy gap further raises to 10.8. This reveals that the iterative optimization mechanism can further improve the quality of the GMM classifier, especially for long-tail languages.
>
> **Q3: Adding an experiment that downgrades the similarity ranking to binary.**
>
> Please refer to the response to Q3 included in the “**General responses to all Reviewers**” part for details.
>
> **Q4: The reason to adopt GMM as classifier.**
>
> Please refer to the response to Q2 included in the “**General responses to all Reviewers**” part for details.

---

> > ### Comment · Reviewer_s5pr · 2023-08-10
> >
> > Thanks for your response. I am looking forward to the experiments that substitutes GMM with a normal classifier, which I think would provide good insights on this design choice during the EM iterations.

---

> > > ### Author Response · Authors · 2023-08-14
> > >
> > > We would like to express our gratitude for your support and encouragement! Your positive feedback means a lot to us, and we are thrilled that you view our work in such a positive light. We will continue to strive for excellence and make further contributions in future research and improvements.

---

### Official Review · Reviewer_FMvk · 2023-07-07

**Soundness:** 3 good
**Presentation:** 3 good
**Contribution:** 2 fair
**Rating:** 6
**Confidence:** 3

**Summary:**

The paper proposes EMMA-X, an EM-like approach to pretrain multilingual models. It learns the cross-lingual representation learning task and semantic relation prediction task within EM. They propose a new benchmark, XRETE, to evaluate the experiments with 12 cross-lingual tasks. The training involves two stages: 1) pretraining using parallel data (bitext) and 2) training with EM Framework with non-parallel data. The proposed approach outperforms the baselines.

**Strengths:**

- The proposed method outperforms other existing pre-trained models, showing the solid multilingual / cross-lingual ability.
- The proposed method is backed by a strong theoretical analysis
- The paper proposes a new benchmark from existing datasets to evaluate the cross-linguality.

**Weaknesses:**

- The main paper does not provide enough information about the model. I would suggest moving the model details from the Appendix to the main paper.
- The benchmark does not provide the model parameters. The comparison may not be fair since the model sizes are different.

**Questions:**

**Suggestion**
- Figure 1 is not easily understandable. It would be great if there is a clear step-by-step procedure in the caption.


**Limitations:**

No. The authors can have a section to address the limitations.

---

> ### Author Rebuttal · Authors · 2023-08-08
>
> Thanks for your positive reviews.
>
> **Q1: Moving model details from Appendix to Main paper.**
>
> Due to page limitations, certain model details have been included in Appendix A. We will incorporate the model details to the main paper in the next version.
>
> **Q2: Lack of information about model parameters in the benchmark.**
>
> We have provided the details of our baseline model in Appendix D and will move this part to main pages in the revision. Accully, most of our baseline models share identical model parameters with EMMA-X, as they are also further trained based on the XLM-R [1] model. The only exception is LaBSE [2], which is further trained based on the BERT model, with the only difference being the size of the vocabularies.
>
> **Q3: Adding a step-by-step procedure in the caption of Figure 1.**
>
> In Section 3.3, we have already introduced EMMA-X step-by-step, and further illustrated the procedure through Algorithm 1. To provide readers with a better understanding of EMMA-X, we will include a step-by-step procedure in the caption of Figure 1.
>
> **References:**
>
> [1] Conneau A, Khandelwal K, Goyal N, et al. Unsupervised cross-lingual representation learning at scale.
>
> [2] Feng F, Yang Y, Cer D, et al. Language-agnostic BERT sentence embedding

---

### Official Review · Reviewer_LAMb · 2023-07-07

**Soundness:** 3 good
**Presentation:** 3 good
**Contribution:** 3 good
**Rating:** 6
**Confidence:** 4

**Summary:**

This paper proposes to apply the EM framework to realize unified sentence representation learning with non-parallel multilingual data. This framework consists of two modules, a GMM classifier and a cross-language encoder, which are responsible for semantically related classification and cross-language unified representation. However, this is achieved on the premise that the parallel multilingual data initializes the models. In addition, the author conducted a theoretical analysis and forms a new benchmark, based on which the effectiveness of the framework was proved.

**Strengths:**

The paper applies the EM algorithm to realize the optimization of the multilingual unified representation parallel-trained model under non-parallel data.

**Weaknesses:**

1.	Why choose the GMM model as the Semantic Relations model?
2.	Why form a XRETE benchmark, what are its innovations and necessity?
3.	This article introduces monolingual data from a continuous perspective, so what are its advantages compared with discrete methods(e.g. Back-translation)


**Questions:**

1.	Why choose the GMM model as the Semantic Relations model?
2.	Why form a XRETE benchmark, what are its innovations and necessity?
3.	This article introduces monolingual data from a continuous perspective, so what are its advantages compared with discrete methods(e.g. Back-translation)

---

> ### Author Rebuttal · Authors · 2023-08-08
>
> Thanks for your positive comments.
>
> **Q1: The reason to use GMM model as Semantic Relation Model.**
>
> Please refer to the response to Q2 included in the “**General responses to all Reviewers**” part for details.
>
> **Q2: The reason to form an XRETE benchmark.**
>
> The formation of the XRETE benchmark is driven by the following reasons:
> 1. Inclusion of recent, dedicated, and challenging benchmarks: XRETE incorporates more up-to-date benchmarks that specifically focus on universal sentence representations. These newly added benchmarks, such as Legal topics in MultiEURLEX [1], Review topics in Multilingual Amazon Review corpus [2], WMT Quality Estimation [3], and Americas NLI [4], offer increased difficulty in evaluating the performance of models.
> 2. Coverage of more low-resource languages: XRETE goes beyond its predecessors by encompassing a broader range of low-resource languages. For instance, it includes 10 Indigenous languages of the Americas, which were not part of the XTREME [5] or XGLUE [6] benchmarks. This expansion ensures that the XRETE benchmark provides a more comprehensive evaluation of models' capabilities across diverse linguistic backgrounds.
>
> **Q3: Advantages of EMMA-X compared with Back-Translation.**
>
> EMMA-X offers several advantages over discrete methods like back-translation:
> 1. Faster parallel corpus construction: EMMA-X outperforms back-translation in constructing parallel corpora at a faster pace. The process of generating supervision signals in EMMA-X involves predicting the semantic relation rank between two sentences, which is a more efficient approach compared to back-translation's auto-regressive generation of pseudo-parallel sentences.
> 2. Higher quality parallel corpora on low-resource languages: In the context of low-resource languages, EMMA-X excels in constructing parallel corpora with higher quality compared to back-translation. Back-translation may yield erroneous results for some languages, or unavailable for some languages such as pa, ps, and others. Conversely, EMMA-X retrieves real sentences from a monolingual, low-resource corpus, ensuring the production of more reliable and accurate parallel data.
> 3. Justification of semantic relation rank predictions during pre-training: EMMA-X uniquely unifies the prediction of semantic relation rank with the learning process of universal semantic representations within an Expectation-Maximization (EM) framework, which allows for the justification of semantic relation rank predictions during pre-training. In contrast, discrete methods like back-translation often rely on fixed translation models for pseudo-parallel corpus construction, leading to limited improvement in sentence representations' quality.
>
> **References:**
>
> [1] Chalkidis I, Fergadiotis M, Androutsopoulos I. MultiEURLEX--A multi-lingual and multi-label legal document classification dataset for zero-shot cross-lingual transfer.
>
> [2] Phillip Keung, Yichao Lu, György Szarvas, and Noah A. Smith. The multilingual Amazon reviews corpus.
>
> [3] Lucia Specia, Frédéric Blain, Marina Fomicheva, Chrysoula Zerva, Zhenhao Li, Vishrav Chaudhary, and André F. T. Martins. Findings of the WMT 2021 shared task on quality estimation.
>
> [4] Ebrahimi A, Mager M, Oncevay A, et al. Americasnli: Evaluating zero-shot natural language understanding of pretrained multilingual models in truly low-resource languages.
>
> [5] Hu J, Ruder S, Siddhant A, et al. Xtreme: A massively multilingual multi-task benchmark for evaluating cross-lingual generalization.
>
> [6] Liang Y, Duan N, Gong Y, et al. XGLUE: A new benchmark dataset for cross-lingual pre-training, understanding and generation.

---

### Author Rebuttal · Authors · 2023-08-08

**General Response to all Reviewers:**

Thank all reviewers for their time and efforts.

**Q1: Clarification about “Unsupervised” / “without parallel data” in EMMA-X.**

In EMMA-X, we explicitly show the use of parallel data for initializing the model in Section 3.2 and Algorithm 1. The primary goal of EMMA-X is to acquire universal semantic representations for a multitude of languages. However, it is important to note that only a limited number of languages (4%) possess parallel data. Therefore, EMMA-X strives to leverage non-parallel data to extend language coverage for universal semantic representations and to enhance representation performance for those languages that have limited available resources.
To further prove our claims, we provided an extra ablation study to show how each part in EMMA-X can affect performance and the results are shown in Table 5.

**Q2: The reason to use GMM classifier.**

Given the absence of direct supervision signals, we intuitively hypothesize that each semantic rank follows a Gaussian distribution. To effectively cluster each sentence pair into its corresponding semantic rank, we employ Gaussian Mixture Model (GMM) as the classifier. Moreover, the GMM classifier offers several advantages:
1. Addressing imbalanced data: The GMM classifier is particularly advantageous in handling imbalanced datasets. It can model clusters of varying sizes, ensuring that minority class data points are not overshadowed by the majority class. By allowing data points to contribute to multiple clusters with different probabilities, the GMM classifier mitigates the imbalance issue effectively.
2. Effective handling of outliers: With its soft assignment approach, the GMM classifier can effectively deal with outliers. Outliers are assigned low probabilities of belonging to any cluster, reducing their impact on shifting the cluster centers or covariances significantly.

To further substantiate our claims, we plan to include an additional experiment that compares the performance of the GMM classifier with that of a normal classifier in the forthcoming version.

**Q3: Choice of semantic similarity rank.**

The selection of the semantic similarity rank was made heuristically, and we chose a rank of 4 based on the following two reasons:
1. Addressing data imbalance: A fine-grained similarity rank allows for a smoother distribution of semantic ranks. In the case of binary classification of semantic relations, there tends to be a significant imbalance between negative (non-semantically similar) and positive samples. By employing a fine-grained semantic rank, we can better distribute sentence pairs originally classified as negatives into different ranks, thus mitigating the data imbalance issue.
2. Learning complexity consideration: Conversely, defining a higher number of semantic ranks poses challenges for both the GMM classifier and the cross-lingual encoder to learn. Therefore, we heuristically arrived at the choice of N=4 as it represents a good trade-off between addressing the data imbalance problem and managing learning complexity.

To provide further support for our approach, we are currently conducting experiments on varying the number of semantic ranks, and we intend to include these findings in the next version of our paper.

---

### Decision · Program_Chairs · 2023-09-21

**Decision:**

Accept (poster)

**Comment:**

This is a solid and well-written paper. Experiments are extensive and show that the proposed approach is effective and gives sota results. The authors should ensure they address all of the reviewers' concerns in the camera ready, in particular the comments about claims (ie, use of parallel data to bootstrap the model); the reasons behind choosing GMMs; the choice of ranking vs a binary approach and choice of semantic ranks; clarification wrt supervised vs unsupervised vs semi-supervised and the appropriateness of (and differences between) baselines and methods. The authors promise to add certain additional experiments in the next version of the paper and they are highly encouraged to do so in the camera ready, as well as Table 5 presented here.